# GRADUAL DOMAIN ADAPTATION VIA GRADIENT FLOW

**Zhan Zhuang**[1,2]**, Yu Zhang**[1,†]**, Ying Wei**[3,†]
[1]Southern University of Science and Technology, [2]City University of Hong Kong
[3]Nanyang Technological University
12250063@mail.sustech.edu.cn, yu.zhang.ust@gmail.com,
ying.wei@ntu.edu.sg

## ABSTRACT

Domain shift degrades classification models on new data distributions. Conventional unsupervised domain adaptation (UDA) aims to learn features that bridge labeled source and unlabeled target domains. In contrast to feature learning, gradual domain adaptation (GDA) leverages extra continuous intermediate domains with pseudo-labels to boost the source classifier. However, real intermediate domains are sometimes unavailable or ineffective. In this paper, we propose **G**radual Domain Adaptation via **G**radient **F**low (GGF) to generate intermediate domains with preserving labels, thereby enabling us a fine-tuning method for GDA. We employ the Wasserstein gradient flow in Kullback–Leibler divergence to transport samples from the source to the target domain. To simulate the dynamics, we utilize the Langevin algorithm. Since the Langevin algorithm disregards label information and introduces diffusion noise, we introduce classifier-based and sample-based potentials to avoid label switching and dramatic deviations in the sampling process. For the proposed GGF model, we analyze its generalization bound. Experiments on several benchmark datasets demonstrate the superiority of the proposed GGF method compared to state-of-the-art baselines.

## 1 INTRODUCTION

Unsupervised Domain Adaptation (UDA) stands as a fundamental and classical problem in machine learning (Pan & Yang, 2010). Its primary objective revolves around the transfer of the knowledge from a well-trained source domain to a related yet unlabeled target domain, thereby reducing the need for time-consuming manual labeling and data preprocessing.

Early discrepancy-based UDA methods (Borgwardt et al., 2006; Pan et al., 2010; Tzeng et al., 2014; Courty et al., 2014; Long et al., 2015; Ganin & Lempitsky, 2015; Sun & Saenko, 2016; Ganin et al., 2016; Courty et al., 2017; Tzeng et al., 2017) have shown promise in learning domain-invariant feature representations. However, recent studies (Zhao et al., 2019; Chen et al., 2019; Tang & Jia, 2020; Yang et al., 2020) reveal that simply aligning source and target domains likely reduces discriminability. To address this issue, transport-based (Kirchmeyer et al., 2022; Gao et al., 2023; Xiao et al., 2023) and synthetic sample-based (Cui et al., 2020; Xu et al., 2020; Wu et al., 2020; Choi et al., 2020; Dai et al., 2021; Na et al., 2021; Jing et al., 2022; Na et al., 2022) approaches have been developed. Apart from them, Kumar et al. (2020) proposed a learning paradigm called Gradual Domain Adaptation (GDA), which resorts to an extra sequence of continuous unlabeled samples as intermediate domains to adapt the source classifier to the target domain via self-training instead of feature alignment.

However, in many real-world scenarios where only unlabeled data from the target domain are available, constructing appropriate intermediate domains remains an open question (Wang et al., 2022), which transforms the UDA setting into the GDA setting. Prior studies have demonstrated that generating additional intermediate domains through generative models or interpolation algorithms offers viable approaches and improves the learning performance of the target domain. For instance, Sagawa & Hino (2022) utilized Continuous Normalizing Flow (CNF) (Chen et al., 2018) to interpolate between the source and target domains. GIFT (Abnar et al., 2021) and GOAT (He et al., 2023) use optimal transport and linear interpolation within a mini-batch to form geodesic paths. Those methods share a

---

[†]Corresponding authors.

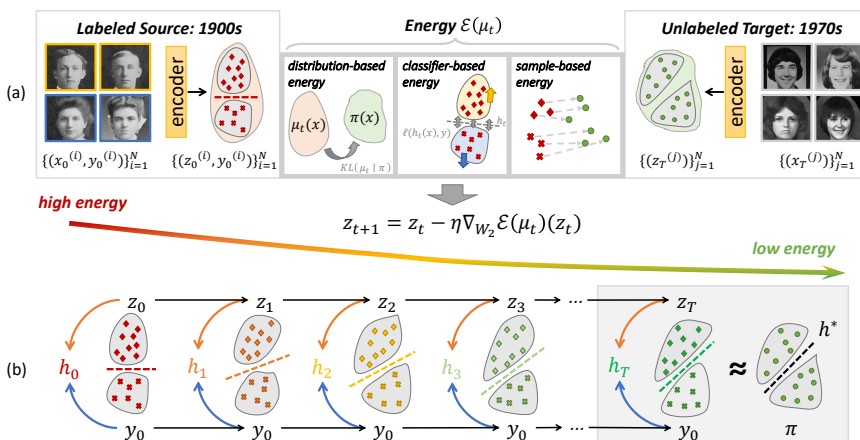

Figure 1: Illustration of the proposed GGF method. (a) The three designed energies collectively define a gradient flow in latent space. (b) The training strategy of GGF involves gradual fine-tuning of the classifier by utilizing the intermediate domains constructed in the GGF method.

similar spirit to transport-based and synthetic sample-based UDA methods. Unfortunately, they suffer from two limitations. Firstly, one-step interpolation and CNFs with minimal constraints can result in ambiguous labels for synthetic samples. Secondly, self-training with multiple pseudo-labeling iterations may lead to training instability (Chen et al., 2022).

To address the limitations above, we propose **G**radual domain adaptation via **G**radient **F**low (GGF), a novel method to construct intermediate domains when intermediate domains are absent. Gradient flow refers to the continuous process of gradient descent, which follows the steepest descent direction to minimize a given energy function. As shown in Figure 1, in the proposed GGF method, the designed gradient flow transports feature representations from the source domain to the target domain along a curve to minimize the following three designed energies: **(1)** *distribution-based energy* that is to shift the features from the source domain to the target domain, **(2)** *classifier-based energy* that is to preserve label information, and **(3)** *sample-based energy* that is to avoid large noise in generated samples. To the best of our knowledge, we are the first to propose the construction of intermediate domains, while meantime allowing the source classifier to be gradually fine-tuned to align with the target distribution. We provide theoretical analysis on the generalization bound of the proposed GGF method, showing that the gradient flow with smaller continuous discriminability and transport loss contributes a lower target error. We conduct comprehensive experiments to evaluate the proposed GGF on various domain adaptation scenarios, where GGF outperforms state-of-the-art methods.

## 2 PRELIMINARIES AND BACKGROUND

### 2.1 SETTING AND NOTATION

In the context of GDA, we consider the data of the labeled source domain, $T$ unlabeled intermediate domains, and the unlabeled target domain to be sampled from the distribution $\mu_0$, $\{\mu_t\}_{t=1}^{T}$ and $\mu_{T+1}$ over $\mathcal{X}$, respectively. We denote the target distribution by $\pi$. Consider the hypothesis class $\mathcal{H}$, where for any classifier $h \in \mathcal{H}, h : \mathcal{X} \to \mathcal{Y}$ maps inputs to predictions. We assume that there exists a labeling function in each domain: $f_{\mu_t} = f_t \in \mathcal{H}$. Given a loss function $\ell(\cdot, \cdot)$, the generalization error is defined as $\epsilon_\mu(h) = \epsilon_\mu(h, f_\mu) = \mathbb{E}_{x \sim \mu} \ell(h(x), f_\mu(x))$. The source classifier $h_0$ can be learned with minimal error $\epsilon_{\mu_0}(h_0)$ using supervised learning, and the objective of GDA is to evolve this classifier $h_0$ to $h_T$ over the intermediate domains so as to minimize the target error $\epsilon_\pi(h_T)$. The UDA problem shares the objective and can be converted into a GDA problem by generating intermediate domains.

**Self-training** Self-training (ST) (French et al., 2017; Zou et al., 2019; Prabhu et al., 2021; Liu et al., 2021) (*a.k.a.* pseudo-labeling) is a domain adaptation method that does not rely on feature learning. We use the Wasserstein-2 distance $W_2(\mu, \nu)$ (see Appendix. A.1) to quantify the domain shift. Under the assumption that the shift between adjacent domains is small, previous studies (Kumar et al., 2020; Wang et al., 2022) have shown that ST can effectively update the classifier $h_{t-1}$ to adapt to the next domain with samples $S_t$, as expressed mathematically below:

$$h_t = \text{ST}(h_{t-1}, S_t) = \arg\min_{h \in \mathcal{H}} \frac{1}{|S_t|} \sum_{\mathbf{x} \in S_t} \ell(h(\mathbf{x}), h_{t-1}(\mathbf{x})), \tag{1}$$

where $h_{t-1}(\mathbf{x})$ denotes pseudo labels of $\mathbf{x}$ by $h_{t-1}$. Kumar et al. (2020) proposed Gradual Self-Training (GST) as the baseline algorithm for GDA, which updates the classifier using ST with pseudo-labels on the intermediate domains in sequence. In this paper, we generate intermediate domains gradually while preserving labels, enabling us to update the classifier in a distinct manner by fine-tuning it directly on the transformed samples.

## 2.2 Wasserstein Gradient Flow (WGF)

A flow describes a time-dependent diffeomorphic map of particles in a system. Let $\Phi_t(\mathbf{x}) : [0, 1] \times \mathbb{R}^n \to \mathbb{R}^n$ denote a flow, so that a vector field of position and time $\mathbf{u}_t(\mathbf{x})$ defines a flow via an ordinary differential equation (ODE), i.e.,

$$\frac{d}{dt}\Phi_t(\mathbf{x}) = \mathbf{u}_t\left(\Phi_t(\mathbf{x})\right), \quad \Phi_0(\mathbf{x}) = \mathbf{x}. \tag{2}$$

Let $\mathcal{P}_2(\mathbb{R}^n)$ denote the space of probability measures on $\mathbb{R}^n$ with finite second moments. For a metric space $(\mathcal{P}_2(\mathbb{R}^n), W_2)$, Otto (2001) first proposed WGF to solve the porous medium equation. The vector field of WGF is defined as $\mathbf{u}_t = -\nabla_{W_2}\mathcal{E}(\mu_t)$, where a general energy $\mathcal{E}(\mu_t)$ consisting of the following three terms associates with a distribution $\mu_t$ or a group of particles (Santambrogio, 2017),

$$\mathcal{E}(\mu_t) = \int H(\mu_t(\mathbf{x}))d\mathbf{x} + \int V(\mathbf{x})d\mu_t(\mathbf{x}) + \frac{1}{2}\iint W(\mathbf{x} - \mathbf{y})d\mu_t(\mathbf{x})d\mu_t(\mathbf{y}). \tag{3}$$

The three terms represent the internal, (external) potential, and interaction energy, respectively. Simply put, internal energy (e.g. entropy $H(\mu) = \mu \log \mu$) is related to distribution density, potential energy is related to the potential field $V$ in Euclidean space, and interaction energy captures the interactions between particles. Correspondingly, the vector field can be calculated as

$$\mathbf{u}_t = -\nabla_{W_2}\mathcal{E}(\mu_t) = -\nabla\mathcal{E}'(\mu_t) = -\nabla(H'(\mu_t)) - \nabla V - (\nabla W) * \mu_t, \tag{4}$$

where $*$ denotes the convolution operator. We can conceptualize the Kullback–Leibler (KL) divergence, maximum mean discrepancy (MMD) (Arbel et al., 2019), or other related metrics (Mroueh & Rigotti, 2020; Korba et al., 2021; Glaser et al., 2021) as the energy functional. Leveraging the gradient flow's descent property, we can establish a progressive flow that reduces the difference between distributions over time. This approach proves valuable for dataset transformation and data augmentation (Alvarez-Melis & Fusi, 2021; Hua et al., 2023). In the next section, we will delve into our proposed method in the context of domain adaptation.

## 3 Gradual Domain Adaptation via Gradient Flow (GGF)

In this section, we present the novel GGF method, designed to construct intermediate domains along the gradient flow of three energy functions and gradually adapt a source classifier to the target domain. We commence by providing a comprehensive exposition of the three energies and their corresponding sampling techniques. Subsequently, we unify these components into a cohesive framework. To allow the versatility and extensive applicability of our approach in both data and latent spaces, we utilize the symbol "$\mathbf{x}$" to represent both samples and features in the subsequent sections.

### 3.1 Distribution-based Energy for Shifting Features from Source to Target

Considering the high computation cost of MMD (Arbel et al., 2019) and KSD (Mroueh & Rigotti, 2020), we opt $\mathrm{KL}(\mu_t|\pi)$ as the distribution-based energy with the resulting vector field given as,

$$\mathbf{u}_t = -\nabla_{W_2}\mathrm{KL}(\mu_t|\pi) = -\nabla_{W_2}\int \log\left(\frac{\mu_t}{\pi}\right)\mathrm{d}\mu_t = -\nabla\log\mu_t + \nabla\log\pi. \tag{5}$$

This vector field consisting of the above two log-density components drives the distribution $\mu_t$ towards the target distribution $\pi$ under the influence of both diffusion and drift (Jordan et al., 1998). We simulate the associated flow following the common practice of Langevin Monte Carlo (LMC) (Roberts & Tweedie, 1996) as a time-discretized sampling method. The LMC guarantees convergence in the Wasserstein distance, as described in Eq. (32) (Dalalyan & Karagulyan, 2019). Concretely, given a set of samples $x_0$ from the source distribution $\mu_0$, the LMC iteratively updates the samples as

$$\mathbf{x}_{t+1} = \mathbf{x}_t + \eta_1\nabla_{\mathbf{x}_t}\log\pi(\mathbf{x}_t) + \sqrt{2\eta_1}\xi, \tag{6}$$

where $\pi$, $\xi$, and $\eta_1$ represent the distribution of the target domain, a noise sampled from a standard Gaussian distribution, and the step size, respectively. The log-density $\nabla_{\mathbf{x}} \log \pi(\mathbf{x})$, also known as the (Stein) score function $s_\pi(\mathbf{x})$, can be estimated via various methods (Hyvärinen & Dayan, 2005; Vincent, 2011; Song et al., 2020), even when the probability density function is not available. Here, we resort to Denoise Score Matching (DSM) (Vincent, 2011) with a neural network $s(\mathbf{x}; \phi)$ to model the score function. The core idea of DSM is to minimize the discrepancy between the score and the model's predicted score after injecting noise to clean $\mathbf{x}$, whose objective follows,

$$J_{DSM} = \mathbb{E}_{q_\sigma(\mathbf{x}, \tilde{\mathbf{x}})} \left[ \frac{1}{2} \| s(\tilde{\mathbf{x}}; \phi) - \nabla_{\tilde{x}} \log q_\sigma(\tilde{\mathbf{x}}|\mathbf{x}) \|^2 \right]. \tag{7}$$

Simple augmentation methods such as Gaussian noise $\epsilon \sim N(\mathbf{0}, \sigma^2 \mathbf{I})$ can be employed to compute the latter term (i.e., $\nabla_{\tilde{x}} \log q_\sigma(\tilde{\mathbf{x}}|\mathbf{x}) = -\frac{\tilde{\mathbf{x}} - \mathbf{x}}{\sigma^2}$), where $\sigma$ denotes the noise variance. By using $q_\sigma(\mathbf{x}, \tilde{\mathbf{x}}) = q_\sigma(\tilde{\mathbf{x}}|\mathbf{x}) p(\mathbf{x})$, we can efficiently estimate the scores of perturbed data $\tilde{\mathbf{x}}$ in those low-density areas of the target distribution. Once the score network has been trained, we can readily construct intermediate domains following the sampling process in Eq. (6).

**Challenges** There still remain two main challenges with the sampled intermediate domains. Firstly, the sampling disregards the label information of the samples, which likely introduces either condition shift $\mathcal{P}_{\mu_0}(y|\mathbf{x}) \neq \mathcal{P}_\pi(y|\mathbf{x})$ or prior shift $\mathcal{P}_{\mu_0}(y) \neq \mathcal{P}_\pi(y)$ between the source and target domains and thereby puts the decision boundary in danger of collapse. Secondly, The noise (*a.k.a.* diffusion) term introduced by LMC may cause some samples to deviate drastically from the true data distribution, given that the score network cannot accurately estimate the shifted points. To address both issues, we propose the following classifier-based and sample-based energy.

## 3.2 CLASSIFIER-BASED ENERGY FOR PRESERVING LABEL INFORMATION

We use the cross-entropy loss and the entropy of the logits as the classifier-based potential energy:

$$\mathcal{L}_{\text{CE}}(\mu, h, y) = -\int y(\mathbf{x}) \log h(\mathbf{x}) d\mu(\mathbf{x}), \quad \mathcal{L}_{\text{H}}(\mu, h) = -\int h(\mathbf{x}) \log h(\mathbf{x}) d\mu(\mathbf{x}), \tag{8}$$

where $y(\mathbf{x})$ denotes the corresponding label of input $\mathbf{x}$, and $h(\mathbf{x})$ denotes its prediction. For the cross-entropy loss, lower energy ensures that the predictions of the shifted samples remain consistent with the original labels. Meanwhile, lower entropy of logits yields higher prediction confidence, avoiding excessively smooth predictions. By introducing $\lambda$, we balance between the two potentials to accommodate different distribution characteristics of datasets. The flow that implements this classifier-based energy is exactly the gradient with respect to inputs via backpropagation, i.e.,

$$\mathbf{x}_{t+1} = \mathbf{x}_t - \eta_2 ((1 - \lambda) \nabla_{\mathbf{x}_t} \mathcal{L}_{\text{CE}}(\mathbf{x}_t, h_t, y_t) + \lambda \nabla_{\mathbf{x}_t} \mathcal{L}_{\text{H}}(\mathbf{x}_t, h_t)). \tag{9}$$

**Observation** Our experiments show that the LMC monotonically reduces the Wasserstein distance between the constructed intermediate domains to the target domain. Including the classifier-based energy, however, results in an initial decrease in the distance to a minimum but followed by an increase, indicating a balance between feature shift and label preserving. We resolve this issue in Section 3.4 by setting a proper number of intermediate domains using this stationary point.

## 3.3 SAMPLE-BASED ENERGY FOR REDUCING NOISE AND RECTIFYING FLOW

To mitigate the noise of generated samples, we borrow the idea from stochastic interpolant (Liu et al., 2023; Liu, 2022; Albergo & Vanden-Eijnden, 2023; Lipman et al., 2023; Albergo et al., 2023) where flows connecting two distributions do not deviate too much. Those methods use interpolation techniques to create interpolants $\mathbf{x}_\tau$, based on which they directly estimate the vector field of the samples through a neural network $v_\theta$ different from WGF that derives a flow from a prior energy. Specifically, we adopt the rectified flow approach (Liu et al., 2023; Liu, 2022) to generate interpolants at time $\tau \in [0, 1]$ as $\mathbf{x}_\tau = (1 - \tau)\mathbf{x}_0 + \tau \mathbf{x}_1$, and subsequently estimates the vector field $v_\theta(\mathbf{x}_\tau)$ with the following flow matching objective:

$$J_{FM} = \mathbb{E}_{\mathbf{x}_0 \sim \mu_0, \mathbf{x}_1 \sim \pi} \left[ \int_0^1 \| v_\theta(\mathbf{x}_\tau) - v(\mathbf{x} | \mathbf{x}_\tau) \|^2 d\tau \right], \tag{10}$$

where the conditional vector field $v(\mathbf{x} \mid \mathbf{x}_\tau) = \dot{\mathbf{x}}_\tau = \mathbf{x}_1 - \mathbf{x}_0$. The estimated vector field $v_\theta(\mathbf{x}_\tau)$ defines a flow, which can be discretized using the Euler method as $\mathbf{x}_{t+1} = \mathbf{x}_t + \eta_3 v_\theta(\mathbf{x}_t)$. Even without explicit prior energy like WGF, this flow guarantees high energy for source samples and low energy for target samples for each interpolation pair.

## 3.4 COMPLETE ALGORITHM

We summarize the three energies above into a unified Wasserstein gradient flow. To balance the three terms, we use step sizes $\eta_1, \eta_2, \eta_3$, and halt the iterated sampling process once the minimum Wasserstein distance to the target domain is reached. We construct each intermediate domain with $\alpha$ iterations, resulting in a total of $\alpha T$ iterations across $T$ intermediate domains. Each iteration follows

$$\mathbf{x}_{t+1} = \mathbf{x}_t \underbrace{+ \eta_1 s_\pi(\mathbf{x}_t; \phi) + \sqrt{2\eta_1}\xi}_{\text{Distribution-based}} \underbrace{- \eta_2 \nabla_{\mathbf{x}_t} \mathcal{L}(\mathbf{x}_t, h_t, y_t)}_{\text{Classifier-based}} \underbrace{+ \eta_3 v_\theta(\mathbf{x}_t)}_{\text{Sample-based}}. \tag{11}$$

A comprehensive algorithm and complexity analysis are described in Appendix C.

## 4 THEORETICAL ANALYSIS

This section provides a theoretical analysis of the proposed GGF method. The primary distinction between our analysis and prior theoretical works (Kumar et al., 2020; Wang et al., 2022) is that we do not use any given intermediate domains. Instead, we design Wasserstein gradient flows to generate the intermediate domains, which incrementally shift the source distribution while preserving labels.

We set the labeling function $f_T$ of the last intermediate and target domains to be the same, which is reasonable considering the synthetic nature of the intermediate domain. Our analysis, with detailed proof in Appendix B, has two main steps: first, we offer an upper-bound of the target error in Lemma 1, using the risk on the last intermediate domain $\epsilon_{\mu_T}(h_T)$ and the final Wasserstein distance $W_2(\mu_T, \pi)$. Then, we derive the two terms and provide a generalization bound in Theorem 1. To facilitate our analysis, we adopt the assumptions consistently with prior works (Dalalyan & Karagulyan, 2019; Kumar et al., 2020; Wang et al., 2022):

**Assumption 1** *Each predictor function $h \in \mathcal{H}$ and labeling function $f \in \mathcal{H}$ is R-Lipschitz in $\ell_2$ norm, i.e., $\forall x, x' \in \mathcal{X} : |h(x) - h(x')| \leq R\|x - x'\|$*

**Assumption 2** *The loss function $\ell$ is $\rho$-Lipschitz, i.e., $\forall y, y' \in \mathcal{Y} : |\ell(y, \cdot) - \ell(y', \cdot)| \leq \rho\|y - y'\|_2$ and $|\ell(\cdot, y) - \ell(\cdot, y')| \leq \rho\|y - y'\|$*

**Assumption 3** *The Rademacher complexity (Bartlett & Mendelson, 2002) of hypothesis class $\mathcal{H}$ denoted by $\Re_n(\mathcal{H})$ is bounded by $\frac{B}{\sqrt{n}}$, i.e., $\Re_n(\mathcal{H}) \leq \frac{B}{\sqrt{n}}$*

**Assumption 4** *The potential $V$ is m-strongly convex, and M-Lipschitz smooth, i.e., $\forall x, x' \in \mathcal{X}$ :*

1. $V(x) \leq V(x') - \langle \nabla V(x), x' - x \rangle - \frac{m}{2}\|x - x'\|^2$    (*m-strongly convex*)

2. $V(x) \leq V(x') + \langle \nabla V(x), x' - x \rangle + \frac{M}{2}\|x - x'\|^2$    (*M-Lipschitz smooth*)

**Assumption 5** *Consider the vector field $\boldsymbol{u}_t$ is bounded, i.e, $\|\boldsymbol{u}_t\| \leq U$*

With Assumptions 1 and 2 as well as the shared labeling function $f_T$, we present Lemma 1. Despite the seemingly similarity with Lemma 1 in (Wang et al., 2022), we note that this paper considers the Wasserstein distance over $\mathcal{X}$.

**Lemma 1** *For any classifier $h \in \mathcal{H}$, the generalization error on the target domain is bounded by the error on the last generated intermediate domain and the Wasserstein distance:*

$$\epsilon_\pi(h) \leq \epsilon_{\mu_T}(h) + 2\rho R W_1(\mu_T, \pi) \tag{12}$$

Our method gradually refines the source classifier $h_0$ from the source $\mu_0$ to the last intermediate domain $\mu_{T-1}$, enabling iteration analysis. Therefore, we introduce Proposition 1, which states the bounded variance of generalization error after updating a classifier on a domain.

**Proposition 1** (The Stability of GGF) *Consider $\{(x_i, y_i)_{i=1}^n\}$ are i.i.d. samples from a domain with distribution $\mu$, and $h_\mu$ is a classifier. GGF provides a map $\mathcal{T}_\mu^\nu$ that transports $x_i$ to the next domain $\nu$ and updates the classifier to $h_\nu$ by empirical risk minimization (ERM) on the shifted samples as $h_\nu = \arg\min_{h \in \mathcal{H}} \sum_{i=1}^n \ell(\mathcal{T}_\mu^\nu(x_i), y_i)$. Denote the labeling functions of the two domains are $f_\mu$ and $f_\nu$, then, for any $\delta \in (0, 1)$, with probability at least $1 - \delta$, the following bound holds true:*

$$|\epsilon_\mu(h_\mu) - \epsilon_\nu(h_\nu)| \le \rho(\mathbb{E}_{x \sim \mu}|f_\mu(x) - f_\nu(x)| + R\Delta_\mu) + \mathcal{O}\left(\frac{\rho B + \sqrt{\log(1/\delta)}}{\sqrt{n}}\right) \quad (13)$$

*where $\Delta_\mu = \frac{1}{n}\sum_{i=1}^n \left\| x_i - \mathcal{T}_\mu^\nu(x_i)\right\|$ means the average shift distance for $x_i$ on $\mathcal{T}_\mu^\nu$.*

In comparing the stability of the GST algorithm (Wang et al., 2022) with ours, we note that the primary difference lies in the first term. Specifically, the GST algorithm bounds the error difference by the Wasserstein distance $W_p(\mu, \nu)$ over the joint distribution between adjacent domains. For our method, the error is bounded using two parts: 1) an expected difference in labeling functions $\mathbb{E}_{x \sim \mu}|f_\mu(x) - f_\nu(x)|$, bounded by the *Continuous Discriminability defined in B.2* and 2) a transport cost $\Delta_\mu$, bounded by the vector field limit.

**Theorem 1** (Generalization Bound for GGF) *Under Assumptions 1-5, if $h_T$ is the classifier on the last intermediate domain updated by GGF, then for any $\delta \in (0, 1)$, with probability at least $1 - \delta$, the generalization error of $h_T$ on target domain is upper bounded as:*

$$\epsilon_\pi(h_T) \le \epsilon_{\mu_0}(h_0) + \epsilon_{\mu_0}(h_T^\star) + \rho R\left(\eta\alpha TU + 2(1 - \eta m)^{\alpha T}W_2(\mu_0, \pi) + \frac{3.3M\sqrt{\eta p}}{m}\right)$$
$$+ \mathcal{O}\left(\frac{\rho B + \sqrt{\log(1/\delta)}}{\sqrt{n}}T\right) \quad (14)$$

Based on the above results, we derive the generalization bound for GGF in Theorem 1. The first two terms $\epsilon_{\mu_0}(h_0)$ and $\epsilon_{\mu_0}(h_T^\star)$ represent the source generalization errors of the source classifier and the target labeling function. Besides, the $\epsilon_\mu(h_T^\star)$ term can also be replaced with a lower term $\rho K$, representing the continuous discriminability above. The $\eta\alpha TU$ term corresponds to the transport cost of samples across intermediate domains through $\alpha T$ iterations with a step size of $\eta_1$ and abbreviated as $\eta$. Meanwhile, $W_2(\mu_0, \pi)$ denotes the Wasserstein distance between the source and target over $\mathcal{X}$.

**Remark:** Prior theoretical works (Kumar et al., 2020; Wang et al., 2022; Dong et al., 2022) link the generalization bound to the Wasserstein distance over $\mathcal{X} \times \mathcal{Y}$ across given intermediate domains. For the GST algorithm, disregarding the $\mathcal{O}(1/\sqrt{nT})$ term introduced in the online learning framework for improved sample complexity, the bound proposed in (Wang et al., 2022) simplifies to

$$\epsilon_\pi(h_T) \le \epsilon_{\mu_0}(h_0) + \rho\sqrt{R^2 + 1}\sum_{t=1}^T W_p(\mu_{t-1}, \mu_t) + \mathcal{O}\left(\frac{\rho B + \sqrt{\log(1/\delta)}}{\sqrt{n}}T\right), \quad (15)$$

which suggests that the optimal intermediate domains should be along the Wasserstein geodesic path from the source to target domains, which is unfortunately non-trivial to pinpoint on unlabeled domains. Instead, the novel upper bound we establish in Eq. (14) provides a promising and practical error bound for analyzing GDA when intermediate domains are generated along the Wasserstein gradient flow. Moreover, our analysis offers insights into minimizing the upper bound through the optimization of the hyperparameters $\alpha$, $T$, and $\eta$. By fixing two of them, the optimal value for the remaining one is deterministic aligning with the lowest upper bound.

## 5 EXPERIMENTS

In Section 5.2, we evaluate the effectiveness of the proposed GGF on three GDA benchmarks and pre-trained features of the UDA task, Office-Home. The experimental results demonstrate that GGF constructs practical intermediate domains and gradually updates the initial classifier more accurately. In Section 5.3, we conduct the ablation studies for the three kinds of energy, pseudo-labeling, multiple iterations, and the number of intermediate domains.

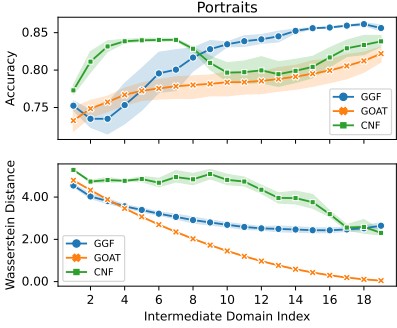

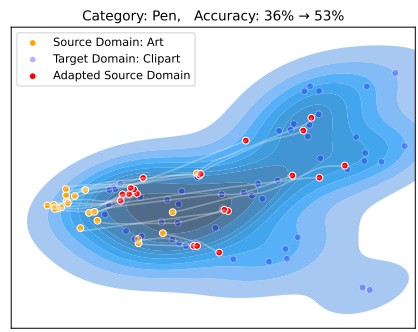

Figure 3: Accuracy (%) and $W_2$ distance to target domain over intermediate domains on Portraits.

Figure 4: The adaptation example of the "pen" category samples from Art to Clipart domain.

To aid understanding, we construct two toy datasets: a mixture of Gaussian blobs and two-moon (details in Appendix D.1). Figure 2 illustrates how GGF transports source samples to the target and updates the classifier.

### 5.1 EXPERIMENTAL SETTINGS

**Datasets** We mainly evaluate our method on five datasets. **Portraits** is a gender (binary) classification dataset with 37,921 facing portraits from 1905 to 2013. We follow the chronological split from Kumar et al. (2020), creating a source domain (first 2000 images), intermediate domains (14000 images not used here), and a target domain (next 2000 images). **Rotated**

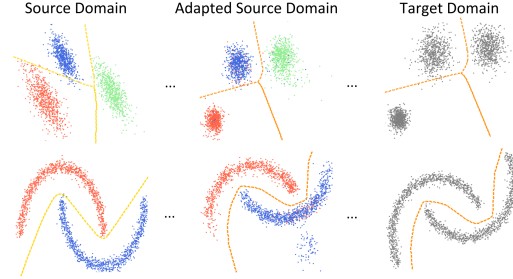

Figure 2: Two toy examples. The dashed lines in the first and other columns correspond to the initial and updated classifiers, respectively.

**MNIST** is a variant of the MNIST dataset (LeCun, 1998) consisting of 4000 source images and 4000 target images rotated by 45° to 60° degrees, as per He et al. (2023) and Kumar et al. (2020). **Office-Home** (Venkateswara et al., 2017) is a well-known UDA dataset with 65 categories across four domains: Artistic (Ar), Clipart (Cl), Product (Pr), and Real-World (Rw). **VisDA-2017** (Peng et al., 2017) is a large-scale dataset including a simulation-to-real UDA task with 152,397 synthetic training images and 72,372 real-world test images across 12 categories.

**Implementation** Following the setting in (Sagawa & Hino, 2022), we use semi-supervised UMAP (McInnes et al., 2018) as the feature extractor to reduce the dimensionality of input data while preserving the class discriminability. For the UDA datasets, we use the extracted features as input for experiments. Further implementation details are deferred to Appendix E.

Table 1: Accuracy (%) on GDA tasks.

|  | Portraits | MNIST 45° | MNIST 60° |
|---|---|---|---|
| Source | 76.50 | 57.06 | 37.47 |
| Self Train | 76.99 | 60.22 | 39.81 |
| GST (4) (Kumar et al., 2020) | 81.45 | 66.45 | 56.87 |
| GOAT (He et al., 2023) | 83.17 | 61.45 | 46.29 |
| CNF (Sagawa & Hino, 2022) | 84.57 | 62.55 | 42.18 |
| GGF (Ours) | **86.16** | **67.72** | **54.21** |

### 5.2 MAIN RESULTS

Results in Table 1 reveal that all three methods can generate efficient intermediate domains for GDA, and the proposed GGF outperforms the others, especially for large domain shifts. Notably, our method is competitive with the baseline GST algorithm with about four real intermediate domains. In Figure 3, we observe the accuracy evolution with the number of intermediate domains on Portraits for each method. GOAT creates samples along the Wasserstein geodesic but has smaller improvements in datasets with small shifts. CNF's fluctuating accuracy indicates some ineffective domains, while our gradient flow-based method consistently reduces domain differences and improves accuracy.

We evaluate GGF on Office-Home tasks and notice that it further improves the accuracy by up to 0.5%, as shown in Table 2. Additionally, we provide visual evidence of source sample adaptation in Figure 4, demonstrating that GGF proficiently transfers source features to the target domain, achieving

Table 2: Classification accuracy (%) on Office-Home dataset with ResNet-50. The best accuracy is indicated in bold, and the second best is underlined.

| Method | Ar→Cl | Ar→Pr | Ar→Rw | Cl→Ar | Cl→Pr | Cl→Rw | Pr→Ar | Pr→Cl | Pr→Rw | Rw→Ar | Rw→Cl | Rw→Pr | Avg. |
|---|---|---|---|---|---|---|---|---|---|---|---|---|---|
| DANN (Ganin & Lempitsky, 2015) | 45.6 | 59.3 | 70.1 | 47.0 | 58.5 | 60.9 | 46.1 | 43.7 | 68.5 | 63.2 | 51.8 | 76.8 | 57.6 |
| MSTN (Xie et al., 2018) | 49.8 | 70.3 | 76.3 | 60.4 | 68.5 | 69.6 | 61.4 | 48.9 | 75.7 | 70.9 | 55.0 | 81.1 | 65.7 |
| GVB-GD (Cui et al., 2020) | 57.0 | 74.7 | 79.8 | 64.6 | 74.1 | 74.6 | 65.2 | 55.1 | 81.0 | 74.6 | 59.7 | 84.3 | 70.4 |
| RSDA (Gu et al., 2020) | 53.2 | 77.7 | 81.3 | 66.4 | 74.0 | 76.5 | 67.9 | 53.0 | 82.0 | 75.8 | 57.8 | 85.4 | 70.9 |
| LAMDA (Le et al., 2021) | 57.2 | 78.4 | 82.6 | 66.1 | 80.2 | 81.2 | 65.6 | 55.1 | 82.8 | 71.6 | 59.2 | 83.9 | 72.0 |
| SENTRY (Prabhu et al., 2021) | 61.8 | 77.4 | 80.1 | 66.3 | 71.6 | 74.7 | 66.8 | 63.0 | 80.9 | 74.0 | 66.3 | 84.1 | 72.2 |
| FixBi (Na et al., 2021) | 58.1 | 77.3 | 80.4 | 67.7 | 79.5 | 78.1 | 65.8 | 57.9 | 81.7 | 76.4 | 62.9 | 86.7 | 72.7 |
| CST (Liu et al., 2021) | 59.0 | 79.6 | 83.4 | 68.4 | 77.1 | 76.7 | 68.9 | 56.4 | 83.0 | 75.3 | 62.2 | 85.1 | 73.0 |
| CoVi (Na et al., 2022) | 58.5 | 78.1 | 80.0 | 68.1 | 80.0 | 77.0 | 66.4 | 60.2 | 82.1 | 76.6 | 63.6 | 86.5 | 73.1 |
| RSDA + GGF | 60.1 | 77.9 | 82.2 | 68.4 | 78.3 | 77.2 | 67.8 | 60.3 | 82.5 | 75.8 | 61.0 | 85.2 | 73.1 |
| Covi + GGF | 59.2 | 79.0 | 80.4 | 69.3 | 80.1 | 78.1 | 66.8 | 61.7 | 83.1 | 76.2 | 62.8 | 86.5 | 73.6 |

superior feature alignment. In Appendix D, we provide GGF's performance on large-scale datasets (D.2), along with a comparative analysis of different feature extractors (D.3) and visualizations (D.5).

## 5.3 ABLATION STUDY

**Different Energy Functions** We conduct an ablation study on the three types of energy in GGF, as outlined in Table 3. Our findings highlight the importance of both the classifier-based and sample-based ener-

Table 3: Ablation study on different energies.

| Distribution-based | Classifier-based | Sample-based | Portraits | Accuracy MNIST 45° | MNIST 60° |
|---|---|---|---|---|---|
| ✓ | | | 82.69 (±0.38) | 66.19 (±0.99) | 46.95 (±0.77) |
| ✓ | ✓ | | 84.02 (±1.42) | 66.88 (±0.56) | 53.15 (±0.73) |
| ✓ | | ✓ | 84.60 (±0.21) | 67.45 (±0.60) | 50.82 (±0.33) |
| ✓ | ✓ | ✓ | 86.16 (±0.19) | 67.72 (±0.34) | 54.21 (±0.86) |

gies in achieving optimal performance with GGF. Notably, the classifier-based energy, functioning as a regularization term, significantly enhances the performance in the MNIST 60° task by preserving category information. This is crucial given the substantial domain shift between the source and target domains, as well as the unsatisfactory predictions of the initial classifier in the target domain.

**Sensitivity Analysis on Step Sizes $\eta$** We perform a sensitivity analysis by rescaling each step size while keeping the other two fixed on Portraits datasets. The results, shown in Table 4, demonstrate the numerical proximity and robustness of our hyperparameters. Notably, we observe a significant decrease in performance only when reduc-

Table 4: Sensitivity analysis on step sizes $\eta$. The default setting of rescale ratio is 100%.

| | 25% | 50% | 75% | 100% | 150% | 200% | 400% |
|---|---|---|---|---|---|---|---|
| $\eta_1 = 0.03$ | 79.00 | 79.15 | 86.45 | 86.35 | 84.85 | 84.75 | 84.10 |
| $\eta_2 = 0.08$ | 84.75 | 85.80 | 86.15 | 86.35 | 74.45 | 78.90 | 68.00 |
| $\eta_3 = 0.01$ | 85.85 | 86.05 | 86.50 | 86.35 | 86.40 | 86.15 | 85.80 |

ing $\eta_1$ or increasing $\eta_2$, indicating that the classifier-based energy dominates the distribution-based energy. This suggests that the former induces larger velocity components, pushing samples away from the decision boundary. Fortunately, our hyperparameter optimization method avoids converging to solutions where the classifier-based energy dominates, as these solutions do not lead to a reduction in the Wasserstein distance from samples to the target domain.

**Comparison of Self-Training (ST) and Fine-Tuning (FT)** Figure 5 compares the two updating methods of GGF: ST with pseudo-labels and FT with preserving labels. In the Portraits task with smaller $\alpha$, ST with a smaller confidence threshold outperforms FT, but as $\alpha$ increases, FT is stable and better. In the MNIST 60° task, ST performs poorly with small $\alpha$ due to cumulative instability. These results demonstrate that ST is less effective with larger domain gaps and more pseudo-labeling processes, highlighting the advantages of our preserving labels approach. Additionally, a sufficiently large $\alpha$ represents the one-step adaptation. As shown in the figures, gradually updating the classifier as in the GDA setting works better than updating the classifier only in the last intermediate domain.

## 6 RELATED WORK

**Unsupervised Domain Adaptation (UDA)** UDA aims to facilitate knowledge transfer from a source domain to a target one with only unlabeled instances. Early UDA methods, divided into the two major categories of discrepancy-based and adversarial-based, center around learning domain-invariant representations. The discrepancy being minimized in discrepancy-based ways includes the Maximum

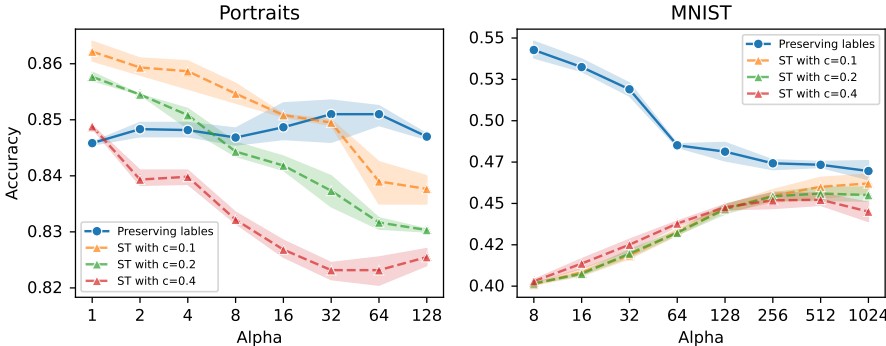

Figure 5: Accuracy comparison of two updating methods with varying hyperparameter $\alpha$ on Portraits and MNIST 60° tasks, using different confidence thresholds $c$ for self-training. With fixed sampling iterations $\alpha T$, the number of intermediate domains $T$ decreases as $\alpha$ increases.

Mean Discrepancy (MMD) (Borgwardt et al., 2006; Long et al., 2015), second-order statistics (Sun & Saenko, 2016), and optimal transport distances (Courty et al., 2014; 2017; Shen et al., 2018; Li et al., 2020; Kerdoncuff et al., 2021). Instead, adversarial-based methods pursue domain-invariant features that are indistinguishable by a domain discriminator (Ganin & Lempitsky, 2015; Tzeng et al., 2017; Long et al., 2018; Liu et al., 2019). Recent research (Chen et al., 2019; Yang et al., 2020; Tang & Jia, 2020) suggests a negative impact on discriminability when dealing with two wildly dissimilar domains, which motivates the transport- and synthetic sample-based methods.

○ *Transport-based methods* preserve domain-specific features by training a feature extractor without transfer loss and then transferring these features from the target to the source domain using optimal transport or sampling techniques. For instance, Kirchmeyer et al. (2022) and Fan & Alvarez-Melis (2023) trained an optimal transport map between conditional distributions in feature space and performs reweighting for label matching and robust pre-training. Recently, Gao et al. (2023) and Xiao et al. (2023) leveraged diffusion and energy-based models to shift target data for test-time input adaptation and preserve the image structure and latent variables related to class information.
○ *Synthetic sample-based methods* create intermediate samples at the input or feature levels by employing techniques such as geodesic flow kernels (Gong et al., 2012), adaptors (Choi et al., 2020), generative models (Hoffman et al., 2018; Gong et al., 2019; Hsu et al., 2020; Cui et al., 2020) or data augmentation (Xu et al., 2020; Wu et al., 2020; Fatras et al., 2022; Dai et al., 2021; Na et al., 2021; Jing et al., 2022; Na et al., 2022; Sahoo et al., 2023). The generated samples are utilized for improved feature learning, consistency regularization, or robust discriminator training.

**Gradual Domain Adaptation (GDA)**     Unlike UDA methods prioritizing feature learning, GDA updates the model using unlabeled sequential intermediate domains, allowing more fine-grained adaptation. Kumar et al. (2020) first introduced the setting with a straightforward gradual self-training (GST) algorithm. Meanwhile, Wang et al. (2020) proposed an adversarial-based method for a similar setting with continuous indexed domains. Based on the assumption of gradually shifting distributions, prior works (Kumar et al., 2020; Dong et al., 2022; Wang et al., 2022) provide and improve the upper generalization bound. Researchers have explored various algorithms for challenging scenarios to obtain the ideal intermediate domains. For instance, when the sequence of extra unlabeled data is unavailable, Chen & Chao (2021) proposed an Intermediate Domain Labeler (IDOL) module to index them. In cases where intermediate domains are insufficient, prior studies have shown that generating additional intermediate domains through continuous normalizing flow (Sagawa & Hino, 2022) or optimal transport and linear interpolation (Abnar et al., 2021; He et al., 2023) can reduce the distance between adjacent domains and improve the performance of GST.

## 7    CONCLUSION

We introduce gradual domain adaptation via gradient flow (GGF), a novel approach to create continuous intermediate domains and incrementally finetune the classifier. Theoretically, We offer an upper-bound analysis of target error. Empirically, we demonstrate that GGF outperforms prior synthetic-based GDA methods and enhances performance compared to state-of-the-art UDA baselines. Our results underscore GGF's ability to generalize effectively across various pre-trained features.

## ACKNOWLEDGEMENTS

This work is supported by NSFC key grant under grant no. 62136005, NSFC general grant under grant no. 62076118, Hong Kong RMGS 9229111, and Shenzhen fundamental research program JCYJ20210324105000003.

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

# A  MATHEMATICAL BACKGROUND

## A.1  WASSERSTEIN METRIC

$\mathcal{P}_2(\mathbb{R}^n)$ denote the space of probability measures on $\mathbb{R}^n$ with finite second moments, i.e. $\mathcal{P}_2(\mathbb{R}^n) = \left\{ \mu \in \mathcal{P}(\mathbb{R}^n), \int \|x\|^2 d\mu(x) < \infty \right\}$. Given $\mu, \nu \in \mathcal{P}_2(\mathbb{R}^n)$, the Wasserstein-$p$ distance between them is defined as

$$W_p(\mu, \nu) = \left( \inf_{\pi \in \Gamma(\mu, \nu)} \int_{\mathbb{R}^n \times \mathbb{R}^n} \|x - y\|^p d\pi(x, y) \right)^{\frac{1}{p}}, \tag{16}$$

where $\Gamma(\mu, \nu)$ is the set of all possible couplings between $\mu$ and $\nu$. For all joint distribution $\pi \in \Gamma(\mu, \nu)$, we have $\mu(x) = \int_{\mathbb{R}^n} \pi(x, y) dy$ and $\nu(y) = \int_{\mathbb{R}^n} \pi(x, y) dx$. The integral on the right-hand side is also considered as the cost of the optimal transport (OT) problem (Kantorovitch's formulation), and the $\pi^*$ is the optimal transport plan. Besides, the monotonicity of the Wasserstein-p distance can be easily shown using Jensen's inequality, which implies that for $1 \leq p \leq q$, $W_p(\mu, \nu) \leq W_q(\mu, \nu)$.

In the measurable space $(\mathcal{P}_2(\mathbb{R}^n), W_2)$ (a.k.a the Wasserstein space), the inner product is defined as:

$$\langle \mu, \nu \rangle_{\mu_t} = \int_{\mathbb{R}^n} \langle \mu(\mathbf{x}), \nu(\mathbf{x}) \rangle_{\mathbb{R}^n} d\mu_t(\mathbf{x}). \tag{17}$$

## A.2  DESCENT PROPERTY OF WASSERSTEIN GRADIENT FLOW

For a measurable space $(\mathcal{P}_2(\mathbb{R}^n), W_2)$, the vector field of a Wasserstein gradient flow is defined as $\mathbf{u}_t = -\nabla_{W_2} \mathcal{E}(\mu_t)$. The descent property is a fundamental property of Wasserstein gradient flow, demonstrating that the rate of change of an energy functional $\mathcal{E}(\mu_t)$ over time is always non-positive along the gradient flow. This property can be mathematically derived using the following equation:

$$\frac{d\mathcal{E}(\mu_t)}{dt} = \left\langle \nabla_{W_2} \mathcal{E}(\mu_t), \frac{d\mu_t}{dt} \right\rangle_{\mu_t} = \langle \nabla_{W_2} \mathcal{E}(\mu_t), -\nabla_{W_2} \mathcal{E}(\mu_t) \rangle_{\mu_t} = -\|\nabla_{W_2} \mathcal{E}(\mu_t)\|_{\mu_t}^2 \leq 0. \tag{18}$$

The first equality is derived using the Chain Rule, while the second equality is obtained by applying the definition of Wasserstein gradient flow.

## A.3  DISCRETIZATION SCHEMES OF WASSERSTEIN GRADIENT FLOW

There are two main discretization methods for Wasserstein Gradient Flow. The forward scheme utilizes gradient descent in the Wasserstein space to determine the steepest movement direction. Given an energy functional $\mathcal{E}(\mu_t)$ and a step size $\gamma$, the forward scheme updates the distribution as follows:

$$\mu_{t+1} = (I - \gamma \nabla_{W_2} \mathcal{E}(\mu_t))_{\#} \mu_t. \tag{19}$$

The backward scheme, also known as the Jordan-Kinderlehrer-Otto (JKO) scheme Jordan et al. (1998), is a well-known discretization method for Wasserstein gradient flow. It involves solving an optimization problem to obtain the updated distribution $\mu_{t+1}$, and is represented as follows:

$$\mu_{t+1} = \operatorname{argmin}_{\mu \in \mathcal{P}_2(\mathbb{R}^d)} \mathcal{E}(\mu) + \frac{1}{2\gamma} W_2^2(\mu, \mu_t). \tag{20}$$

# B  PROOFS

## B.1  PROOF OF LEMMA 1 (ERROR DIFFERENCE OVER LAST INTERMEDIATE AND TARGET DOMAIN)

**Lemma 1** *For any classifier $h \in \mathcal{H}$, the generalization error on the target domain is bounded by the error on the last generated intermediate domain and the Wasserstein distance:*

$$\epsilon_\pi(h) \leq \epsilon_{\mu_T}(h) + 2\rho R W_1(\mu_T, \pi) \tag{12}$$

Note that the generalization error is defined as $\epsilon_\mu(h) = \epsilon_\mu(h, f_\mu) = \mathbb{E}_{x \sim \mu} \ell(h(x), f_\mu(x))$, where $h$ represents the predictor and $f_\mu$ represents the labeling function defined on the data distribution $\mu$.

*Proof.* We start by using the definition of the generalization error to get

$$
\begin{aligned}
|\epsilon_{\mu_T}(h) - \epsilon_\pi(h)| &= |\mathbb{E}_{x \sim \mu_T}[\ell(h(x), f_T(x))] - \mathbb{E}_{x' \sim \pi}[\ell(h(x'), f_T(x'))]| \\
&= \left| \int \ell(h(x), f_T(x)) d\mu_T - \int \ell(h(x'), f_T(x')) d\pi \right| \\
&= \left| \int \ell(h(x), f_T(x)) - \ell(h(x'), f_T(x')) d\gamma \right|,
\end{aligned}
\tag{21}
$$

where $\gamma$ is any coupling (joint distribution) of $\mu_T$ and $\nu$. Then we have

$$
\begin{aligned}
|\epsilon_{\mu_T}(h) - \epsilon_\pi(h)| &\leq \int |\ell(h(x), f_T(x)) - \ell(h(x'), f_T(x'))| d\gamma \\
&\leq \int |\ell(h(x), f_T(x)) - \ell(h(x'), f_T(x))| \\
&\quad + |\ell(h(x'), f_T(x)) - \ell(h(x'), f_T(x'))| d\gamma \\
&\leq \int \rho(\|h(x) - h(x')\| + \|f_T(x) - f_T(x')\|) d\gamma \\
&\leq \int 2\rho R \|x - x'\| d\gamma.
\end{aligned}
\tag{22}
$$

In the above inequalities, we have used the triangle inequality and the Lipschitz continuity property of the loss function $\ell$ and the predictor $h$ (label function $f$) to derive the desired result. After considering the arbitrary coupling $\gamma$, we can use the definition of the Wasserstein distance to get

$$
\begin{aligned}
\epsilon_\pi(h) &\leq \epsilon_{\mu_T}(h) + \inf_\gamma \int 2\rho R \|x - x'\| d\gamma \\
&= \epsilon_{\mu_T}(h) + 2\rho R W_1(\mu_T, \pi) \\
&\leq \epsilon_{\mu_T}(h) + 2\rho R W_2(\mu_T, \pi).
\end{aligned}
\tag{23}
$$

## B.2 DEFINITION OF CONTINUOUS DISCRIMINABILITY

**Definition 1** (Continuous Discriminability) *Let $f_0 = h_0^\star$ and $f_T = h_T^\star$ be the **Bayes optimal predictors** on source and target domain. For R-Lipschitz labelers $\{f_t\}_{t=1}^T$ on the intermediate domains, we can define $f_t$ to satisfy the following minimization objective and denote the minimum as $K$. In particular, we let $K_0$ be the value of $K$ when all labeling functions within the intermediate domains are set to $h_T^\star$, and there exists $K \leq K_0$:*

$$
K = \inf_f \sum_{t=0}^{T-1} \mathbb{E}_{x \sim \mu_t} |f_t(x) - f_{t+1}(x)| \leq \mathbb{E}_{x \sim \mu_0} |h_0^\star(x) - h_T^\star(x)| = K_0
\tag{24}
$$

In this study, only $f_0$ and $f_T$ are determined based on the true distributions of the source and target domains. For theoretical analysis, we need to set appropriate labeling functions $f_t$ for intermediate domains. As stated in Definition 1, we minimize the bound above to specify the labeling function and obtain the minimum value $K$. It is worth noting that $K$ depends on the distribution of the generated intermediate domains. The discriminability of the feature representations can be measured by the term $\lambda = \epsilon_\mu(h^\star) + \epsilon_\nu(h^\star)$, as described in the classic UDA theory (Ben-David et al., 2006). When only one intermediate domain exists, $K$ equals $\lambda$. For more intermediate domains, we can define $K$ to represent the *Continuous Discriminability* of the model across domains. In an ideal scenario, minimizing the value $K$ to zero is possible. Figure 1(b) provides such an example, where $\mathbb{E}_{x \sim \mu_t} |f_t(x) - f_{t+1}(x)| = 0$ for any two adjacent domains, indicating that while two adjacent labeling functions may differ, they perform equally well on the distribution of the former domain. However, since $K$ is non-deterministic, we utilize the deterministic upper bound $K_0$ for further analysis, which corresponds to the source generalization error of the target labeling function $\epsilon_{\mu_0}(f_T)$.

### B.3 PROOF OF PROPOSITION 1 (THE STABILITY OF GGF)

**Proposition 1** (The Stability of GGF) *Consider $\{(x_i, y_i)_{i=1}^n\}$ are i.i.d. samples from a domain with distribution $\mu$, and $h_\mu$ is a classifier. GGF provides a map $\mathcal{T}_\mu^\nu$ that transports $x_i$ to the next domain $\nu$ and updates the classifier to $h_\nu$ by empirical risk minimization (ERM) on the shifted samples as $h_\nu = \arg\min_{h \in \mathcal{H}} \sum_{i=1}^n \ell(\mathcal{T}_\mu^\nu(x_i), y_i)$. Denote the labeling functions of the two domains are $f_\mu$ and $f_\nu$, then, for any $\delta \in (0, 1)$, with probability at least $1 - \delta$, the following bound holds true:*

$$|\epsilon_\mu(h_\mu) - \epsilon_\nu(h_\nu)| \leq \rho(\mathbb{E}_{x \sim \mu}|f_\mu(x) - f_\nu(x)| + R\Delta_\mu) + \mathcal{O}\left(\frac{\rho B + \sqrt{\log(1/\delta)}}{\sqrt{n}}\right) \tag{13}$$

*where $\Delta_\mu = \frac{1}{n} \sum_{i=1}^n \left\| x_i - \mathcal{T}_\mu^\nu(x_i) \right\|$ means the average shift distance for $x_i$ on $\mathcal{T}_\mu^\nu$.*

*Proof.* We first introduce $\epsilon_\mu(h_\mu, f_\nu)$ and use the triangle inequality to split the left-hand side (LHS) into two terms as

$$\begin{aligned}
|\epsilon_\mu(h_\mu) - \epsilon_\nu(h_\nu)| &= |\epsilon_\mu(h_\mu, f_\mu) - \epsilon_\nu(h_\nu, f_\nu)| \\
&\leq |\epsilon_\mu(h_\mu, f_\mu) - \epsilon_\mu(h_\mu, f_\nu)| + |\epsilon_\mu(h_\mu, f_\nu) - \epsilon_\nu(h_\nu, f_\nu)|.
\end{aligned} \tag{25}$$

For the first term, we can express it as the difference in the labeling function:

$$\begin{aligned}
|\epsilon_\mu(h_\mu, f_\mu) - \epsilon_\mu(h_\mu, f_\nu)| &= |\mathbb{E}_{x \sim \mu}\ell(h_\mu(x), f_\mu(x)) - \mathbb{E}_{x' \sim \mu}\ell(h_\mu(x'), f_\nu(x'))| \\
&\leq \mathbb{E}_{x \sim \mu}|\ell(h_\mu(x), f_\mu(x)) - \ell(h_\mu(x), f_\nu(x))| \\
&\leq \rho \mathbb{E}_{x \sim \mu}|f_\mu(x) - f_\nu(x)|.
\end{aligned} \tag{26}$$

For the second term, we introduce the empirical error $\hat{\epsilon}_\mu(h, f_\mu) = \frac{1}{n} \sum_{i=1}^n \ell(h(x_i), f_\mu(x_i))$, where $x_i$ represents the $i$-th sample drawn from the distribution $\mu$. Then, using the triangle inequality, we can then obtain the following:

$$\begin{aligned}
|\epsilon_\mu(h_\mu, f_\nu) - \epsilon_\nu(h_\nu, f_\nu)| \leq {}& |\epsilon_\mu(h_\mu, f_\nu) - \hat{\epsilon}_\mu(h_\mu, f_\nu)| + |\hat{\epsilon}_\mu(h_\mu, f_\nu) - \hat{\epsilon}_\nu(h_\nu, f_\nu)| \\
& + |\hat{\epsilon}_\nu(h_\nu, f_\nu) - \epsilon_\nu(h_\nu, f_\nu)|.
\end{aligned} \tag{27}$$

To bound the difference between the generalization and empirical errors, we utilize the Rademacher complexity (Bartlett & Mendelson, 2002). This bound is also described in Lemma A.1 of the previous work by Kumar et al. (2020). Then, we can get

$$\begin{aligned}
|\epsilon_\mu(h_\mu, f_\nu) - \epsilon_\nu(h_\nu, f_\nu)| &\leq |\hat{\epsilon}_\mu(h_\mu, f_\nu) - \hat{\epsilon}_\nu(h_\nu, f_\nu)| + \mathcal{O}\left(\Re_n(\ell \circ \mathcal{H}) + \sqrt{\frac{\log(1/\delta)}{n}}\right) \\
&\leq |\hat{\epsilon}_\mu(h_\mu, f_\nu) - \hat{\epsilon}_\nu(h_\nu, f_\nu)| + \mathcal{O}\left(\frac{\rho B + \sqrt{\log(1/\delta)}}{\sqrt{n}}\right),
\end{aligned} \tag{28}$$

We can obtain the second inequality using Talagrand's Contraction Lemma (Ledoux & Talagrand, 1991), which bounds the Rademacher complexity of the composition of a hypothesis set $\mathcal{H}$ and a Lipschitz function $\ell$ by $\rho\Re_n(\mathcal{H})$, where $\rho$ is the Lipschitz constant of $\ell$ and $\Re_n(\mathcal{H})$ is the Rademacher complexity of the hypothesis set $\mathcal{H}$ concerning the given dataset.

We rely on the fine-tuning classifier updating method to obtain the first term of the inequality in Eq. (28). This approach involves updating the classifier using the transported samples with preserving

labels. Specifically, we have $h_\mu(x_i) = h_\nu \circ \mathcal{T}_\mu^\nu(x_i)$. Then, we can get

$$
\begin{aligned}
|\hat{\epsilon}_\mu(h_\mu, f_\nu) - \hat{\epsilon}_\nu(h_\nu, f_\nu)| &= \left| \frac{1}{n} \sum_{i=1}^n \ell(h_\mu(x_i), f_\nu(x_i)) - \frac{1}{n} \sum_{i=1}^n \ell(h_\nu(\mathcal{T}_\mu^\nu(x_i)), f_\nu(\mathcal{T}_\mu^\nu(x_i))) \right| \\
\text{(Triangle inequality)} &\le \frac{1}{n} \sum_{i=1}^n \left| \ell(h_\mu(x_i), f_\nu(x_i)) - \ell(h_\nu \circ \mathcal{T}_\mu^\nu(x_i), f_\nu \circ \mathcal{T}_\mu^\nu(x_i)) \right| \\
\text{(Triangle inequality)} &\le \frac{1}{n} \sum_{i=1}^n \left| \ell(h_\mu(x_i), f_\nu(x_i)) - \ell(h_\nu \circ \mathcal{T}_\mu^\nu(x_i), f_\nu(x_i)) \right| \\
&\quad + \frac{1}{n} \sum_{i=1}^n \left| \ell(h_\nu \circ \mathcal{T}_\mu^\nu(x_i), f_\nu(x_i)) - \ell(h_\nu \circ \mathcal{T}_\mu^\nu(x_i), f_\nu \circ \mathcal{T}_\mu^\nu(x_i)) \right| \\
\text{(Using } h_\mu(x_i) = h_\nu \circ \mathcal{T}_\mu^\nu(x_i)) &= \frac{1}{n} \sum_{i=1}^n \left| \ell(h_\nu \circ \mathcal{T}_\mu^\nu(x_i), f_\nu(x_i)) - \ell(h_\nu \circ \mathcal{T}_\mu^\nu(x_i), f_\nu \circ \mathcal{T}(x_i)) \right| \\
\text{(Using } \rho\text{-Lipschitz)} &\le \frac{1}{n} \sum_{i=1}^n \rho \left| f_\nu(x_i) - f_\nu \circ \mathcal{T}_\mu^\nu(x_i) \right| \\
\text{(Using } R\text{-Lipschitz)} &\le \frac{1}{n} \sum_{i=1}^n \rho R \left\| x_i - \mathcal{T}_\mu^\nu(x_i) \right\| \\
\text{(Definition of } \Delta_\mu) &= \rho R \Delta_\mu \\
\text{(Limitation of } \mathbf{u}_t) &\le \rho R \eta \alpha U
\end{aligned}
\tag{29}
$$

By plugging Eqs. (26)-(29) into Eq. (25), we reach the conclusion.

### B.4 Proof of Theorem 1

**Theorem 1** (Generalization Bound for GGF) *Under Assumptions 1-5, if $h_T$ is the classifier on the last intermediate domain updated by GGF, then for any $\delta \in (0,1)$, with probability at least $1 - \delta$, the generalization error of $h_T$ on target domain is upper bounded as:*

$$
\begin{aligned}
\epsilon_\pi(h_T) &\le \epsilon_{\mu_0}(h_0) + \epsilon_{\mu_0}(h_T^\star) + \rho R \left( \eta \alpha T U + 2(1 - \eta m)^{\alpha T} W_2(\mu_0, \pi) + \frac{3.3 M \sqrt{\eta p}}{m} \right) \\
&\quad + \mathcal{O}\left( \frac{\rho B + \sqrt{\log(1/\delta)}}{\sqrt{n}} T \right)
\end{aligned}
\tag{14}
$$

*Proof.* In Lemma 1, we have the inequality $\epsilon_\pi(h_T) \le \epsilon_{\mu_T}(h_T) + 2\rho R W_2(\mu_T, \pi)$. To prove this inequality further, we will bound the first term $\epsilon_{\mu_T}(h_T)$ using the accumulation of Proposition 1, and bound the second term $W_2(\mu_T, \pi)$ based on the nature of the gradient flow.

We can apply Proposition 1 recursively to get

$$
\begin{aligned}
|\epsilon_{\mu_0}(h_0) - \epsilon_{\mu_T}(h_T)| &\le \sum_{t=1}^{t=T} \left| \epsilon_{\mu_{t-1}}(h_{t-1}) - \epsilon_{\mu_t}(h_t) \right| \\
&\le \sum_{t=1}^{t=T} \mathbb{E}_{x \sim \mu_{t-1}} \left| \ell(h_{t-1}(x), f_{t-1}(x)) - \ell(h_{t-1}(x), f_t(x)) \right| + \rho R \eta \alpha T U + \mathcal{O}\left( \frac{\rho B + \sqrt{\log(1/\delta)}}{\sqrt{n}} T \right).
\end{aligned}
\tag{30}
$$

Only the label functions $f_0$ and $f_T$ are determined based on the true distributions of the source and target domains. To select the optimal label functions $f_t$ for the intermediate domains, we minimize the first term of the right-hand side of Eq. (30), similar to the definition of $K$ and $K_0$. Therefore, for

a simple strategy for assigning all labeling functions $f_t$ on the intermediate domains be $f_T$, we have[1]

$$
\begin{aligned}
&\sum_{t=1}^{t=T} \mathbb{E}_{x \sim \mu_{t-1}} \left| \ell\left(h_{t-1}(x), f_{t-1}(x)\right) - \ell\left(h_{t-1}(x), f_t(x)\right) \right| \\
&\leq \mathbb{E}_{x \sim \mu_0} \left| \ell\left(h_0(x), f_0(x)\right) - \ell\left(h_0(x), f_T(x)\right) \right| \\
&= \mathbb{E}_{x \sim \mu_0} \ell\left(h_0(x), f_T(x)\right) - \mathbb{E}_{x \sim \mu_0} \ell\left(h_0(x), f_0(x)\right) \\
&\leq \mathbb{E}_{x \sim \mu_0} \ell\left(h_0(x), f_0(x)\right) + \mathbb{E}_{x \sim \mu_0} \ell\left(f_0(x), f_T(x)\right) - \mathbb{E}_{x \sim \mu_0} \ell\left(h_0(x), f_0(x)\right) \\
&= \mathbb{E}_{x \sim \mu_0} \ell\left(f_0(x), f_T(x)\right) = \epsilon_{\mu_0}(f_T) = \epsilon_{\mu_0}(h_T^\star).
\end{aligned}
\tag{31}
$$

To bound $W_2(\mu_T, \pi)$, we use the convergence properties of the LMC in the Wasserstein metric, which have been investigated extensively (Durmus & Moulines, 2019; Dalalyan, 2017; Cheng & Bartlett, 2018; Dalalyan & Karagulyan, 2019; Durmus et al., 2019; Vempala & Wibisono, 2019). Suppose the potential is $m$-strongly convex and $M$-Lipschitz smooth. Then by the Theorem 1 from Dalalyan & Karagulyan (2019), we have

$$
W_2(\mu_T, \pi) \leq (1 - m\eta)^{\alpha T} W_2(\mu_0, \pi) + 1.65(M/m)\sqrt{\eta p},
\tag{32}
$$

where $\eta \leq \frac{2}{m+M}$ is the constant step-size, and $p$ is the dimension of samples. This theorem shows the LMC implies exponential convergence in the Wasserstein distance. Moreover, replacing the LMC with the JKO scheme (Jordan et al., 1998) would further eliminate the second term (Salim et al., 2020).

Finally, by combining Eq. (30)-(32), we have

$$
\begin{aligned}
\epsilon_\pi(h_T) &\leq \epsilon_{\mu_T}(h_T) + 2\rho R W_2(\mu_T, \pi) \\
&\leq \epsilon_{\mu_0}(h_0) + \epsilon_{\mu_0}(h_T^\star) + \rho R \eta \alpha T U + \mathcal{O}\left(\frac{\rho B + \sqrt{\log(1/\delta)}}{\sqrt{n}} T\right) + 2\rho R W_2(\mu_T, \pi) \\
&\leq \epsilon_{\mu_0}(h_0) + \epsilon_{\mu_0}(h_T^\star) + \rho R \left(\eta \alpha T U + 2(1 - \eta m)^{\alpha T} W_2(\mu_0, \pi) + \frac{3.3 M \sqrt{\eta p}}{m}\right) \\
&\quad + \mathcal{O}\left(\frac{\rho B + \sqrt{\log(1/\delta)}}{\sqrt{n}} T\right),
\end{aligned}
\tag{33}
$$

where we conclude.

## C COMPLETE ALGORITHM AND COMPLEXITY ANALYSIS

The process to construct intermediate domains is outlined in Algorithm 1. Assuming that $N$ represents the size of the source domain dataset, the space and time complexities are $\mathcal{O}(N)$ and $\mathcal{O}(\alpha N)$, respectively. In practice, we divide the dataset into multiple batches, keeping consistent complexities.

---

**Algorithm 1:** Construct Next Intermediate Domain $(\mathbf{x}_t, y_t, s_{\pi,\phi}, h_t, v_\theta, \alpha, \eta)$

---

**Input:** Samples $(\mathbf{x}_t, y_t)$, Classifier $h_t$, Score network $s_\pi(\mathbf{x}_t; \phi)$, Rectified flow $v_\theta(\mathbf{x})$,
       Hyperparameters $\alpha, \eta, \lambda$
**Output:** Adapted samples $(\mathbf{x}_{t+1}, y_{t+1})$
\# Simplify the classifier-based potential
$\mathcal{L}(\mathbf{x}_t, h_t, y) = (1 - \lambda)\mathcal{L}_{\mathrm{CE}}(\mathbf{x}_t, h_t, y) + \lambda \mathcal{L}_{\mathrm{H}}(\mathbf{x}_t, h_t)$
\# Update samples using three energy functions and construct each domain after $\alpha$ iterations
**repeat $\alpha$ times**
    $\mathbf{x}_t \leftarrow \mathbf{x}_t \underbrace{- \eta_1 s_\pi(\mathbf{x}_t; \phi) + \sqrt{2\eta_1}\xi}_{Distribution-based} \underbrace{- \eta_2 \nabla_{\mathbf{x}_t} \mathcal{L}(\mathbf{x}_t, h_t, y_t)}_{Classifier-based} \underbrace{+ \eta_3 v_\theta(\mathbf{x}_t)}_{Sample-based}$
**end**
$(\mathbf{x}_{t+1}, y_{t+1}) \leftarrow (\mathbf{x}_t, y_t)$

---

[1] We can use $\rho K$ to bound this, where $K$ is denoted as $\inf \sum_{t=0}^{T-1} \mathbb{E}_{x \sim \mu_t} |f_t(x) - f_{t+1}(x)|$. $K$ is able to represent the *Continuous Discriminability* of the intermediate domains. However, since $K$ is non-deterministic, we analyze the deterministic upper bound $\epsilon_{\mu_0}(h_T^\star)$.

---

**Algorithm 2:** Complete Algorithm with Fixed Hyperparameters

---

**Input:** Source samples $(\mathbf{x}_{src}, y_{src})$, target samples $\mathbf{x}_{tgt}$, expected number of intermediate
        domains $T$, fixed hyperparameters $\eta$ and $\alpha$, learning rate $k$, evaluation frequency $K$
**Output:** Target classifier $h_T$
Train initial classifier $h_0$ according to cross entropy loss $\mathcal{L}_{CE}$ on source samples;
Train score network $s_{\pi,\phi}$ for distribution-based energy according to $J_{DSM}$;
Train rectified flow $v_\theta$ for sample-based energy according to $J_{FM}$;
$t \leftarrow 0, \mathbf{x}_0 \leftarrow \mathbf{x}_{src}, dis \leftarrow W_2(\mathbf{x}_0, \mathbf{x}_{tgt})$;
# OR operator is a short-circuit operator
**while** $t \bmod K \neq 0$ *OR* $W_2(\mathbf{x}_t, \mathbf{x}_{tgt}) \leq dis$ **do**
    **if** $t \bmod K = 0$ **then**
        |   $dis \leftarrow W_2(\mathbf{x}_t, \mathbf{x}_{tgt})$;
    **end**
    $(\mathbf{x}_{t+1}, y_{t+1}) \leftarrow$ Construct Next Intermediate Domain $(\mathbf{x}_t, y_t, s_{\pi,\phi}, h_t, v_\theta, \alpha, \eta)$;
    $h_{t+1} \leftarrow \arg\min_{h \in \mathcal{H}} \ell(\mathbf{x}_{t+1}, y_{t+1})$;
    $t \leftarrow t + 1$;
**end**

---

**Algorithm 3:** Complete Algorithm with Bilevel Optimization

---

**Input:** Source samples $(\mathbf{x}_{src}, y_{src})$, target samples $\mathbf{x}_{tgt}$, expected number of intermediate
        domains $T$, initialized hyperparameters $\eta$ and $\alpha$, learning rate $k$
**Output:** Target classifier $h_T$
Train initial classifier $h_0$, score network $s_{\pi,\phi}$ and rectified flow $v_\theta$.
**for** *epoch* $\leftarrow 0$ **to** *MaxEpoch* **do**
    $t \leftarrow 0, \mathbf{x}_0 \leftarrow \mathbf{x}_{src}, dis \leftarrow W_2(\mathbf{x}_0, \mathbf{x}_{tgt})$;
    # Fix $\eta$, update $\alpha$
    **while** $t \bmod K \neq 0$ *OR* $W_2(\mathbf{x}_t, \mathbf{x}_{tgt}) \leq dis$ **do**
        **if** $t \bmod K = 0$ **then**
            |   $dis \leftarrow W_2(\mathbf{x}_t, \mathbf{x}_{tgt})$;
        **end**
        $(\mathbf{x}_{t+1}, y_{t+1}) \leftarrow$ Construct Next Intermediate Domain $(\mathbf{x}_t, y_t, s_{\pi,\phi}, h_t, v_\theta, \alpha, \eta)$;
        $h_{t+1} \leftarrow \arg\min_{h \in \mathcal{H}} \ell(\mathbf{x}_{t+1}, y_{t+1})$;
        $t \leftarrow t + 1$;
    **end**
    Update $\alpha$ as $\alpha \leftarrow \lfloor \frac{\alpha t}{T} \rfloor$;
    # Fix $\alpha$, update $\eta$
    **for** $t \leftarrow 0$ **to** $T$ **do**
        $(\mathbf{x}_{t+1}, y_{t+1}) \leftarrow$ Construct Next Intermediate Domain $(\mathbf{x}_t, y_t, s_{\pi,\phi}, h_t, v_\theta, \alpha, \eta)$;
        $h_{t+1} \leftarrow \arg\min_{h \in \mathcal{H}} \ell(\mathbf{x}_{t+1}, y_{t+1})$;
    **end**
    Update $\eta$ as $\eta \leftarrow k\nabla_\eta W_2(\mathbf{x}_T, \mathbf{x}_{tgt})$;
**end**

---

Algorithm 2 provides a comprehensive overview of our complete algorithm. This approach dynamically determines the optimal number of iterations based on the Wasserstein distance within a mini-batch. To compute this distance, we employ the Sinkhorn solver from the toolbox (Flamary et al., 2021), which has demonstrated nearly $\mathcal{O}(n^2)$ space and time complexities, where $n$ signifies the batch size. To further alleviate the computational burden associated with distance calculation, we introduce an evaluation frequency parameter $K$. Consequently, the space and time complexities become $\mathcal{O}(N + n^2)$ and $\mathcal{O}(\alpha TN + Tn^2/K)$, respectively. If we fix the values for the hyper-parameter $T$, the space and time complexities reduce to $\mathcal{O}(N)$ and $\mathcal{O}(\alpha TN)$, respectively.

Due to the monotonicity property, our analysis in Eq. (14) suggests that a hyper-parameter with the lowest upper bound remains deterministic when others are held constant. We introduce a bi-level optimization approach, detailed in Algorithm 3, to find appropriate hyper-parameter settings. In our experiments, we apply this optimization exclusively to the GDA dataset. We do not perform hyperparameter optimization for the Office-Home and VisDA-2017 datasets due to gradient vanishing issues caused by numerous iterations. Instead, we use grid search to determine the hyper-parameters.

# D    ADDITIONAL EXPERIMENTAL DETAILS AND RESULTS

## D.1    TOY DATASET

Figure 6 illustrates the two toy datasets. The first dataset is a mixture of three Gaussian blobs (3 classes), while the second dataset is the two-moon dataset (2 classes) implemented using scikit-learn Pedregosa et al. (2011). Both datasets consist of 1000 samples per class in the source domain. To construct the target domain, we sample an additional 1000 samples from the same distribution and then shift the samples. We change the mean and standard variance for the Gaussian blob dataset to create a shifted distribution. We rotated samples around the origin by 45 degrees for the two-moon dataset.

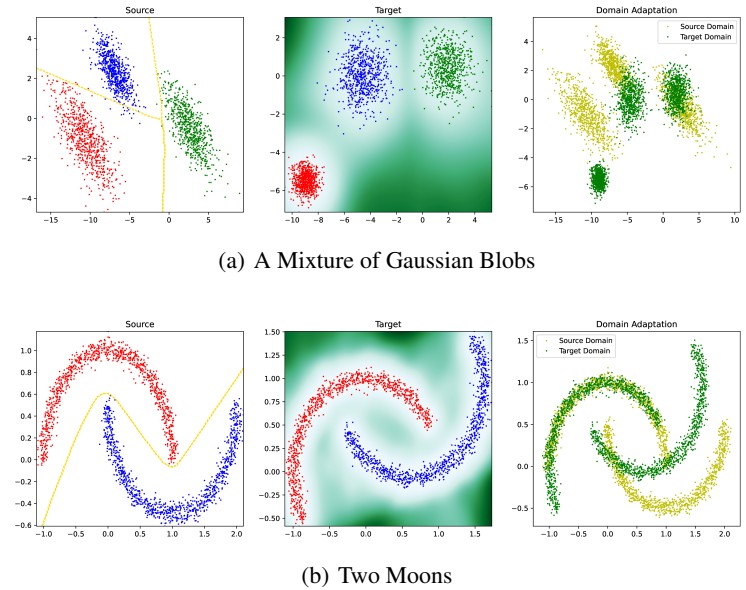

(a) A Mixture of Gaussian Blobs

(b) Two Moons

Figure 6: The left figures show the source domain data and initial classifier, the middle ones show the target domain data and estimated probability density by denoise score matching, and the right ones show an intuitive comparison between the source and target domains.

## D.2    EVALUATION ON THE LARGE-SCALE DATASET

To further validate the effectiveness of our algorithm, we conducted an additional experiment on this dataset. We utilized the state-of-the-art RSDA method for feature extraction and applied GGF in the feature space. As shown in Tables 5 and 6, the results demonstrate notable improvements, with an average accuracy increase of 2.3% across classes and a 1.8% enhancement in overall accuracy.

Table 5: Accuracy (%) on VisDA-2017 (ResNet-50).

| Method | Synthetic → Real |
|---|---|
| CDAN (Long et al., 2018) | 70.0 |
| MDD (Zhang et al., 2019) | 74.6 |
| RSDA (Gu et al., 2020) | 75.8 |
| RSDA + GGF | **77.6** |

Table 6: The detailed accuracy (%) on VisDA campared with RSDA.

| Method | aero | bicycle | bus | car | horse | knife | motor | person | plant | skate | train | truck | Avg. | Acc. |
|---|---|---|---|---|---|---|---|---|---|---|---|---|---|---|
| RSDA | 92.4 | 87.3 | 88.9 | 74.0 | 94.2 | 0.06 | 89.1 | 44.8 | 93.4 | 93.1 | 84.5 | 42.4 | 73.72 | 75.75 |
| RSDA + GGF | 92.4 | 85.2 | 87.1 | 73.3 | 94.1 | 0.07 | 89.7 | 70.0 | 94.0 | 90.6 | 84.5 | 43.9 | 75.98 | 77.56 |

## D.3 COMPARATIVE EVALUATION OF VARIOUS FEATURE EXTRACTORS

Table 7: Accuracy (%) on Office-Home dataset with ResNet-50 (Red for increased accuracy).

| Method | Ar→Cl | Ar→Pr | Ar→Rw | Cl→Ar | Cl→Pr | Cl→Rw | Pr→Ar | Pr→Cl | Pr→Rw | Rw→Ar | Rw→Cl | Rw→Pr | Avg. |
|---|---|---|---|---|---|---|---|---|---|---|---|---|---|
| ResNet-50 (fine-tuned) | 46.2 | 71.5 | 74.0 | 58.4 | 68.1 | 69.7 | 55.8 | 42.3 | 72.9 | 66.9 | 47.3 | 76.0 | 62.4 |
| ResNet-50 + GGF | 48.5 | 74.0 | 76.7 | 61.2 | 70.0 | 71.4 | 60.3 | 44.2 | 75.4 | 69.1 | 49.5 | 77.9 | 64.9 |
| Δ for ResNet-50 | +2.3 | +2.5 | +2.7 | +2.8 | +1.9 | +1.7 | +4.5 | +1.9 | +2.5 | +2.2 | +2.2 | +1.9 | +2.5 |
| MSTN | 57.5 | 70.9 | 77.0 | 60.4 | 71.0 | 69.2 | 61.4 | 56.3 | 79.6 | 70.9 | 54.4 | 80.4 | 67.4 |
| MSTN + GGF | 58.1 | 75.9 | 79.7 | 66.5 | 75.7 | 75.1 | 65.6 | 58.6 | 81.5 | 74.3 | 60.0 | 84.2 | 71.3 |
| Δ for MSTN | +0.6 | +5.0 | +2.7 | +6.1 | +4.7 | +5.9 | +4.2 | +2.3 | +1.9 | +3.4 | +5.6 | +4.2 | +3.9 |
| RSDA | 53.2 | 77.7 | 81.3 | 66.4 | 74.0 | 76.5 | 67.9 | 53.0 | 82.0 | 75.8 | 57.8 | 85.4 | 70.9 |
| RSDA + GGF | 60.1 | 77.9 | 82.2 | 68.4 | 78.3 | 77.2 | 67.8 | 60.3 | 82.5 | 75.8 | 61.0 | 85.2 | 73.1 |
| Δ for RSDA | +6.9 | +0.2 | +0.9 | +2.0 | +4.3 | +0.7 | -0.1 | +6.7 | +0.5 | 0 | +3.2 | -0.2 | +2.2 |
| CoVi | 58.5 | 78.1 | 80.0 | 68.1 | 80.0 | 77.0 | 66.4 | 60.2 | 82.1 | 76.6 | 63.6 | 86.5 | 73.1 |
| CoVi + GGF | 59.2 | 79.0 | 80.4 | 69.3 | 80.1 | 78.1 | 66.8 | 61.7 | 83.1 | 76.2 | 62.8 | 86.5 | 73.6 |
| Δ for CoVi | +0.7 | +0.9 | +0.4 | +1.2 | -0.1 | +0.9 | +0.4 | +1.4 | +1.0 | -0.4 | -0.9 | 0 | +0.5 |

In Table 7, we reiterate the performance of our method on the Office-Home dataset with three different feature extractors. Our primary objective and contribution revolve around the generalization of pre-trained features by any feature extractor, which is readily accessible nowadays, to target domains. Through the comprehensive empirical evaluation, we provide compelling evidence to demonstrate the consistent superiority of GGF even when considering pre-trained features generated by CoVi, which has already shown the SOTA performance in reducing the distribution shift.

## D.4 COMPUTATIONAL OVERHEAD ANALYSIS

In this section, we detail the computational overhead associated with our approach. The training process of GGF can be broadly divided into two stages. The first stage involves training the score network and rectified flow, with the time required contingent upon the number of training epochs. The second stage entails the generation of intermediate domains and the subsequent gradual fine-tuning of the classifier. GGF discretizes

Table 8: Running time (s) on GDA tasks.

| Datasets | GOAT | CNF | GGF (Ours) |
|---|---|---|---|
| Portraits | **2.84** | 5.25 | 5.76 |
| MNIST 45° | **7.95** | 14.23 | 29.53 |

the Wasserstein gradient flow using a forward scheme, which leads to a time complexity that is linearly dependent on the number of intermediate domains and the scale of the dataset. During this stage, our time consumption for generating intermediate domains and updating the classifier is commensurate with those of the baseline methods, GOAT and CNF. Table 8 presents the running time for each method on the two GDA datasets.

## D.5 VISUALIZATION

To demonstrate the process of generating intermediate domains for the proposed GGF method, we visualize the evolution of source samples using t-SNE on the Portraits, and MNIST 45° in Figures 7-8, respectively. We equally generated a small number of intermediate domains and selected them for visualization. These figures show that the source samples evolve over the intermediate domains, indicating that the gradual domain adaptation algorithm effectively adapt the source distribution to the target distribution.

In the UDA task on Office-Home, which involves 65 categories, we conduct a detailed analysis of the t-SNE visualizations for specific classes. As shown in Figures 9 and 10, we visualize the categories where the performance can be improved (success cases) and degraded (failure cases).

In the success cases, we observe that same-class samples in the source domain tend to cluster together with slight variance, while their corresponding target domains are more widely distributed and do not group as a single cluster. Additionally, the intermediate domains generated by GGF can better cover the target distribution. In the failure cases, we observe that the performance degradation is mainly due to the possible diffusion of source domain samples to other similar categories. For instance, in Figure 10(c), the source samples in the "Bottle" category shift in the opposite direction of the target domain distribution, resulting in a 6% decrease in accuracy for this category. However, all those samples are classified into a similar category "Soda".

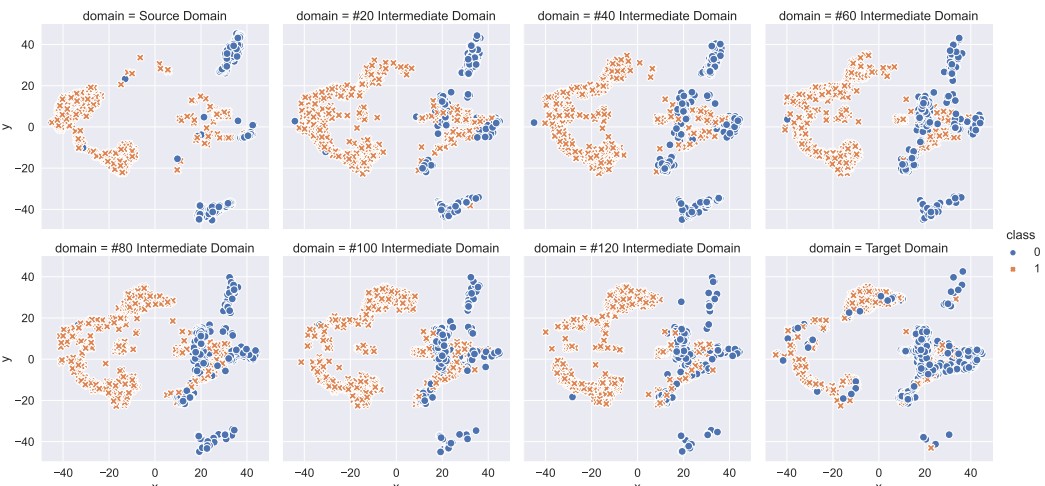

Figure 7: The t-SNE of features from the source, intermediate, and target domains on Portraits.

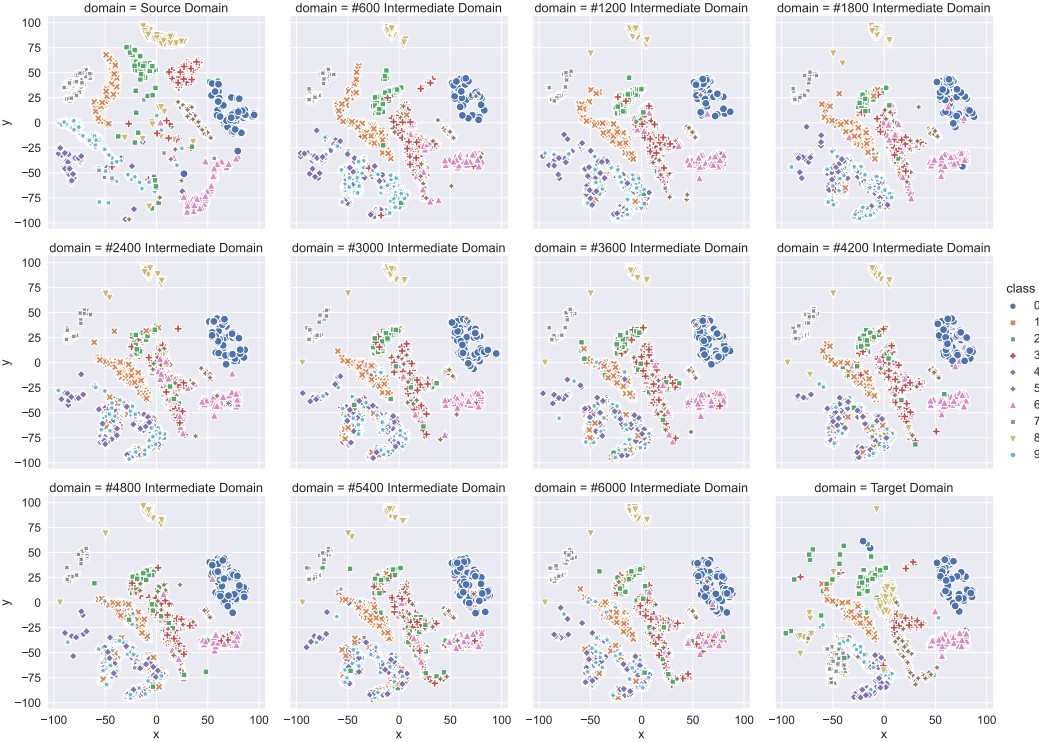

Figure 8: The t-SNE of features from the source, intermediate, and target domains on MNIST 45°.

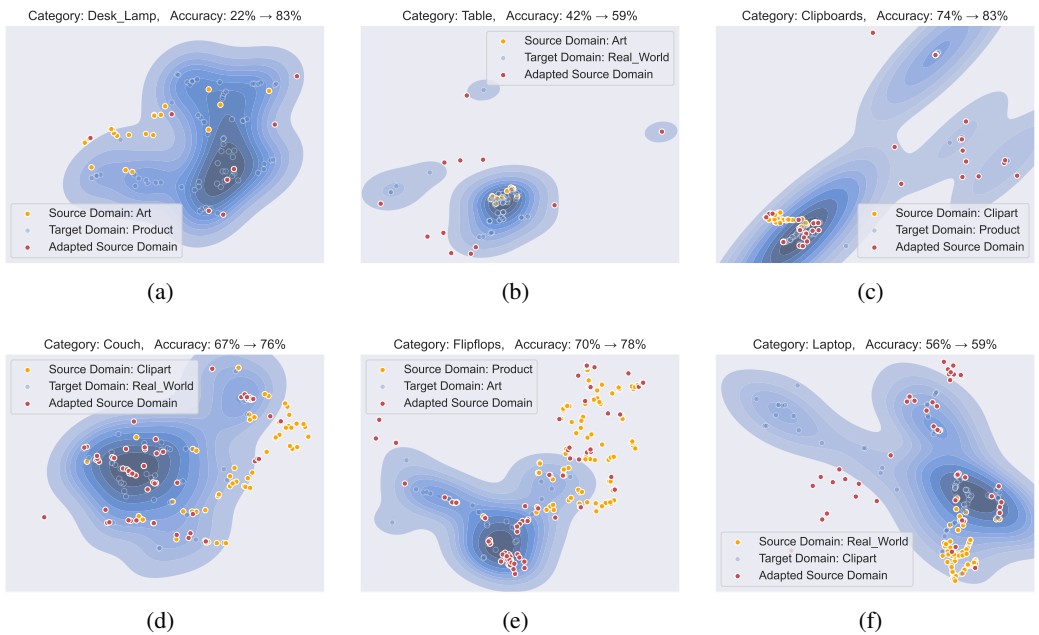

Figure 9: Visualization of success cases, with source samples (in yellow), the last shifted intermediate samples (in red), and target samples (in blue). Each subfigure corresponds to a category in a single task and displays the corresponding accuracy change for that category.

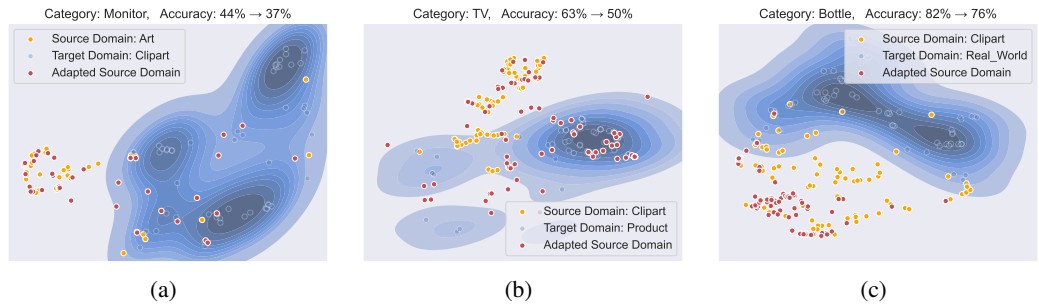

Figure 10: Visualization of failure cases, and the settings are the same as Figure 9.

To further validate the effectiveness of our generated intermediate domains, we present visual representations depicting the relationship between the transported examples and the pre-existing examples in the intermediate domains of two GDA datasets. These visualizations on Portraits and rotated MNIST are depicted in Figure 11 and Figure 12, respectively. In these figures, ellipses represent the variance of samples, while transparency indicates the domain index (with the source domain being the most transparent and the target domain being the least transparent). Each intermediate domain is represented solely by its mean and variance within these visualizations.

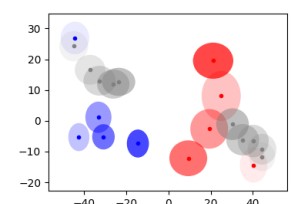

Figure 11: T-SNE visualization of generated (grey) and existing intermediate domains (colorful) on Portraits.

For Portraits, due to the imperfect alignment of facial features over the years, previous work (Chen & Chao, 2021) has shown limited overall improvement when using existing intermediate domains. As shown, while our transported examples lie along a straight trajectory in latent space from source to target, they do not precisely match the real intermediate examples. By generating smoother transport, our method achieves more stable accuracy improvements and even surpasses the accuracy of using the existing examples. For rotated MNIST, many categories perfectly align our transported examples and the existing examples in the t-SNE visualizations. This indicates our model successfully learned the "rotation" features. However, for some categories, the alignment is still poorer, leading to lower performance than gradual self-training on the real intermediate samples.

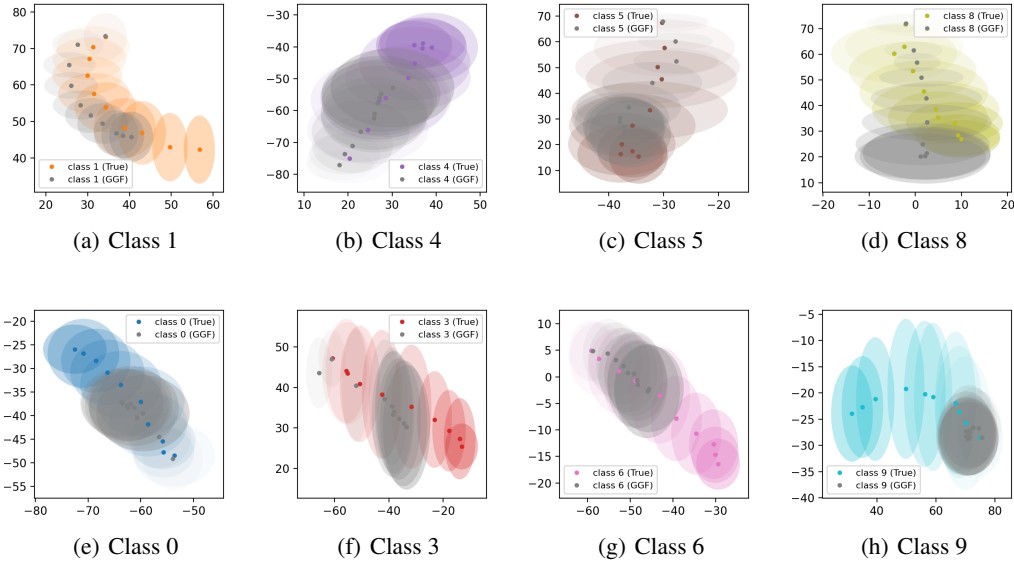

Figure 12: T-SNE visualization of generated (grey) and existing domains (colorful) on rotated MNIST. First row: categories well-aligned with real domains. Second row: categories with fewer alignments.

In summary, our generated examples are comparable or even superior to the existing intermediate samples, effectively bridging the gap between the source and target domains.

Table 9: Hyperparameters of GGF on different datasets.

| Dataset | Confidence Threshold | $\lambda$ | $\alpha$ | $T$ | $\eta_1$ | $\eta_2$ | $\eta_3$ |
|---|---|---|---|---|---|---|---|
| Portraits | 0.05 | 0 | 10 | 20 | 0.03 | 0.08 | 0.01 |
| MNIST 45° | 0.2 | 1 | 100 | 60 | 0.01 | 0.005 | 0.002 |
| MNIST 60° | n/a | 0.8 | 10 | 500 | 0.1 | 0.02 | 0.005 |
| Ar→Cl | n/a | 1 | 10 | 10 | 0.1 | 0.05 | 0.001 |
| Ar→Pr | n/a | 1 | 10 | 20 | 0.03 | 0.01 | 0.001 |
| Ar→Rw | n/a | 1 | 10 | 20 | 0.05 | 0.02 | 0.0001 |
| Cl→Ar | n/a | 1 | 10 | 20 | 0.01 | 0.01 | 0.001 |
| Cl→Pr | n/a | 1 | 10 | 20 | 0.01 | 0.001 | 0.001 |
| Cl→Rw | n/a | 1 | 10 | 20 | 0.001 | 0.001 | 0.001 |
| Pr→Ar | n/a | 1 | 10 | 20 | 0.03 | 0.06 | 0.001 |
| Pr→Cl | n/a | 1 | 100 | 5 | 0.2 | 0.05 | 0.001 |
| Pr→Rw | n/a | 1 | 100 | 20 | 0.005 | 0.005 | 0.0001 |
| Rw→Ar | n/a | 1 | 10 | 50 | 0.2 | 0.2 | 0.001 |
| Rw→Cl | n/a | 1 | 10 | 10 | 0.1 | 0.1 | 0.001 |
| Rw→Pr | n/a | 1 | 10 | 10 | 0.01 | 0.01 | 0.001 |

## E    IMPLEMENTATION

### E.1    TRAINING SETTINGS

We use the official implementations[2][3] for the GOAT (He et al., 2023) and CNF (Sagawa & Hino, 2022) methods and use UMAP (McInnes et al., 2018) to reduce the dimensions of three GDA datasets to 8. We conduct experiments on a single NVIDIA 2080Ti GPU. In the context of the two UDA datasets, namely Office-Home and VisDA-2017, we re-implement the RSDA (Gu et al., 2020) and CoVi (Na et al., 2022) as feature extractors on a single NVIDIA V100 GPU and then conduct our methods on the latent space similarly. For the Rectified Flow (Liu et al., 2023; Liu, 2022), we adopt the official implementation[4]. Our code is publicly available at https://github.com/zwebzone/ggf.

### E.2    HYPERPARAMETERS

In our experiments, we train three neural networks (i.e., score network, rectified flow, and classifier), each consisting of three fully connected layers. For the classifier, we train it in two steps: training the initial classifier and then updating it using the intermediate domains. We apply SGD optimizer with a learning rate of $10^{-4}$ for training all modules and updating the classifier. The batch size for each domain is set to 1024. To improve the training of the score network and rectified flow, we weigh the target domain samples using the classification confidence of the initial classifier.

During the gradual adaptation phase, we update the classifier in each intermediate domain using self-training or fine-tuning

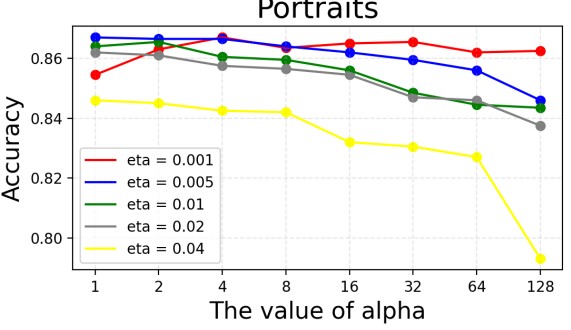

Figure 13: Accuracy comparison with varying hyperparameters on Portraits.

with five epochs. For self-training, we set a confidence threshold $c$ to filter out the least confident examples, which means that $c * 100\%$ of the samples will be removed when updating the classifier. For fine-tuning, we use all generated samples with preserving labels and note that the confidence threshold $c$ is not applicable in this context.

We have discussed how to optimize hyperparameters in Appendix C. Here, we provide the hyperparameters in Table 9, where $\lambda$ is the balancing weight between the cross-entropy and entropy in the class-based energy function, and $\alpha$, $T$, and $\eta$ are used in the sampling process. As shown in Figure 13, our method is not particularly sensitive to those hyperparameters when setting $\alpha$ as a small integer.

[2]https://github.com/yifei-he/GOAT
[3]https://github.com/ssgw320/gdacnf
[4]https://github.com/gnobitab/RectifiedFlow

