# OpenReview forum: "Gradual Domain Adaptation via Gradient Flow"
_ICLR.cc/2024/Conference — ICLR 2024 spotlight_

### Official Review · Reviewer_T69j · 2023-10-29

**Soundness:** 3 good
**Presentation:** 3 good
**Contribution:** 3 good
**Rating:** 6
**Confidence:** 4

**Summary:**

The manuscript introduces a novel approach for generative gradual domain adaptation, termed Gradual Domain Adaptation via Gradient Flow (GGF). This method produces intermediate domain data and iteratively adapts classifiers through a sequence of these intermediate domains. The core concept involves creating intermediate domains between the source and target domains by optimizing a Wasserstein gradient flow to reduce distribution discrepancies. The gradient flow specifically aims to minimize three energies (which the Wasserstein gradient flow is decomposed into):
i) a distribution-related energy for aligning source features with the target
ii) a classifier-related energy to maintain label integrity
iii) a sample-related energy to attenuate noise.

This facilitates the incremental adaptation of the source classifier to the target domain via fine-tuning processes. The authors offer a theoretical framework to constrain the target error based on the properties of the gradient flow. The efficacy of GGF is supported through extensive experiments on standard Domain Adaptation benchmarks.

**Strengths:**

+ [**Novelty**] The idea of connecting gradual domain adaptation (GDA) and Wasserstein gradient flow is natural, given that previous works have already developed theories of GDA with Wasserstein distance. However, the decomposition of Wasserstein gradient flow into three energy terms by [Santambrogio, 2017] is less known in the machine learning community. The authors managed to apply this decomposition to the GDA problem in a non-trivial way: they relate each energy term with some machine learning notion, and find proper ML loss terms as proxies for these energy terms.

+ [**Theoretical Guarantee**] The theoretical analysis grounds the approach well by providing insight into properties like transport cost and label preservation that are useful for analyzing GDA. Bounding the target error connects the gradient flow construction to actual adaptation performance. This helps justify design choices made in GGF.

+ [**Extensive Experiments**] The method is general and can work with different base classifiers and DA techniques that provide feature representations. Comprehensive experiments on benchmark datasets including rotated MNIST, Portraits and Office-Home demonstrate clear improvements over prior arts. The consistent gains across tasks highlight the effectiveness of GGF.

+ [**Visualization**] I quite appreciate the visualizations in Fig. 1, 2, 4, which illustrate the proposed method clearly.

**Weaknesses:**

+ [**No Explanation of the Original Energy Decomposition**] The algorithm is motivated by [Santambrogio, 2017]'s decomposition of Wasserstein gradient flow into three energy terms. However, the authors only describe them as "internal, (external) potential, and interaction energy" without demonstrating their specific definitions. It would be more convincing if the authors can list the original definitions of the energy terms and compare the deviation of their designed loss terms from the original terms.

+ [**Too Many Hyperparameters** ] The loss has three terms, each with a learning rate; there is also an `alpha` term and a `T` term as hyper-parameters. Overall, I feel the number of hyper-parameters is a little high, which may makes the algorithm harder to apply in practice.

+ [**Number of Intermediate Domains should be varied for fair comparison**] In Sec. 5.2, the authors show the improvement of their proposed GGF over GOAT & CNF on the Portraits dataset in Fig. 3. However, the number of intermediate domains in Fig. 3 is fixed to 19. For the GOAT algorithm, its authors demonstrate that the optimal number of intermediate domains for GOAT is below 5, and more intermediate domains may harm its performance. The current choice of 19 intermediate domains might be optimal for GGF, but not for GOAT. So I urge the authors to compare GGF vs. GOAT/CNF across various numbers of intermediate domains (e.g., 2, 4, 8, 16).

**Questions:**

+ [**UDA Experiment Lacks Explanation**] The experiment on Office-Home (shown in Table 2) shows "RSDA+GGF" and "CoVi+GGF" without explanation. I know CoVi also (implicitly) generates intermediate domains. How do you apply GGF on top of these methods? The paper lacks a detailed explanation of your experimental protocols here.

---

> ### Author Response · Authors · 2023-11-19
> **Response to Reviewer T69j (1/2)**
>
> Thank you for your positive feedback and acknowledgment of our novelty. We have addressed your questions point by point in our response below. Please let us know if our explanation adequately answers your inquiries about our paper.
>
> #### [W.1] No explanation of the original energy decomposition.
>
> > - Explanation of original definitions of the energy terms: Consider an energy functional in the Wasserstein space, which characterizes the energy associated with a distribution or a group of particles. Eq. (3) decomposes this general energy into three terms:
> >
> >    - The **internal energy** $\int H(\mu(\mathbf{x})) d \mathbf{x}$ is related to the integral of **a function $H$ of the probability density $\mu$** across all positions in space. For instance, entropy $H(\mu) = \mu \log \mu$ can serve as internal energy, inducing particle Brownian motion (heat flow).
> >    - The **potential energy** $\int V(\mathbf{x}) d \mu(\mathbf{x})$ corresponds to the sum over the potentials of all particles in the distribution with $V(\mathbf{x})$ representing **the potential of each particle $\mathbf{x}$** in Euclidean space.
> >    - The **interaction energy** $\frac{1}{2}\iint W(\mathbf{x} - \mathbf{y}) d \mu(\mathbf{x}) d \mu(\mathbf{y})$ captures interactions between different particles of the distribution, such as gravitational or repulsive forces.
> >
> > - Connections of the three specific energies proposed in our work to the aforementioned general energies:
> >    - The **distribution-based energy** quantifies a distance between the distribution of the $t$-th intermediate domain $\mu_t$ and the target distribution $\pi$, such as $\text{KL}(\mu_t, \pi)$ we consider here. Previous work [r1] has shown that the KL divergence can be expressed as **a combination of** internal energy $H(\mu_t) = \mu_t \log \mu_t$ and potential energy $V(x) = - \log \pi(x)$.
>     >    - The **classifier loss** $\mathcal{L}\left(x, h, y\right)$ is a function of $x$, which serves as potential energy $V(x) =  \mathcal{L}\left(x, h, y\right)$. For cross-entropy loss, lower energies for correct classifications help preserve the conditional distribution $P(y|x)$.
> >    - The **sample-based energy**, derived from the flow matching loss in Eq.(10), generates higher energies for target samples compared to source samples. This can be interpreted as an (interaction) potential, where source samples are attracted by target samples.
> >
> > - To improve clarity, we have included an explanation of the energy decomposition in Section 2.2 in the revised manuscript.
> >
> > [r1] Maximum Mean Discrepancy Gradient Flow, NeurIPS 2019.
>
> #### [W.2] Too many hyperparameters, which may make the algorithm harder to apply in practice.
>
> > We would like to underline humbly that our theoretical results combined with a comprehensive sensitivity analysis facilitate the selection of the 5 hyperparameters, including the $\eta_1$, $\eta_2$, $\eta_3$ that balance three energies, $\alpha$ that controls the frequency of fine-tuning, and $T$ as the number of intermediate domains.
> > - As stated at the end of Section 4, the proposed **Theorem 1 offers insights into hyperparameter selection** for $\eta_1$, $\alpha$, and $T$. Due to the monotonicity of the upper bound, the optimal value for the
> remaining hyperparameter after fixing two of them is deterministic, which aligns with the lowest upper bound.
> >
> > - Our hyperparameter sensitivity analysis with respect to
> >   - $T$ in Figure 3 shows that as long as $T$ is above a value of 10, the performance is quite stable;
> >   - $\alpha$ in Figures 5 and 13 shows that our method is not sensitive to variations in $\alpha$ when it is set as a small integer;
> >   - $\eta_1$, $\eta_2$, $\eta_3$ in the newly included Table 4 demonstrate that they are quite insensitive in a wide range.

---

> ### Author Response · Authors · 2023-11-19
> **Response to Reviewer T69j (2/2)**
>
> #### [W.3] The number of intermediate domains should be varied for a fair comparison.
> >  We have followed the reviewer's insightful suggestion to vary the number of intermediate domains ($T$), exploring values from [2,4,8,16]. We compare the performance of ours with the baselines of GOAT and CNF. The results are presented in both the table below and also our revised paper, with the maximum value along each row highlighted in **bold** and the best performance for a dataset indicated in <font color=red>**red**</font>. We conclude that
> >  - There is a consistent trend across all methods, revealing that **performance tends to improve with an increased number of intermediate domains**. Though the GOAT's paper reports that the optimal number of intermediate domains is less than 5 and the corresponding performances on Portraits and MNIST 45° are 74.2\% and 50.3\%, respectively, our implementations, adhering to the dimensionality reduction practice and network architecture outlined in the CNF's paper, not only elevate GOAT's best performances to 83.7% and 62.2%, but also render the optimal achieved at larger numbers of intermediate domains.
> >  - The current choice of 20 intermediate domains is **fair for both GOAT and CNF**, considering that the optima of either GOAT or CNF are obtained at 16/32 intermediate domains.
> >
> >     | Dataset; Method | 2     | 4     | 8         | 16        | 32        |
> >     | --------------- | ----- | ----- | --------- | --------- | --------- |
> >     | Portraits; GOAT | 80.00 | 81.53 | 83.03     | **83.71** | 82.97     |
> >     | Portraits; CNF  | 84.83 | 81.50 | 84.15     | **84.85** | 79.67     |
> >     | Portraits; GGF  | 85.60 | 85.95 | 86.20     | 86.40     | <font color=red>**86.65**</font> |
> >     | MNIST 45°; GOAT | 59.76 | 61.20 | 61.43     | **62.15** | 62.07     |
> >     | MNIST 45°; CNF  | 60.68 | 61.78 | 62.31     | 62.62     | **62.75** |
> >     | MNIST 45°; GGF  | 58.77 | 59.72 | 62.42     | 64.05     | <font color=red>**66.30**</font> |
> >     | MNIST 60°; GOAT | 41.19 | 41.95 | 43.22     | 44.50     | **46.05** |
> >     | MNIST 60°; CNF  | 41.83 | 42.73 | **43.18** | 43.13     | 42.45     |
> >     | MNIST 60°; GGF  | 42.10 | 46.12 | 47.08     | 48.93     | <font color=red>**51.95**</font> |
>
>
>
>
> #### [Q.1] The UDA experiment lacks an explanation about how to apply GGF on top of UDA methods.
>
> > As detailed in the experimental implementation (Section 5.1), our proposed framework strategically **leverages pre-extracted features as inputs**, including **either** those generated by UDA methods such as RSDA, CoVi, and MSTN, **or** those pre-trained features like the ImageNet pre-trained ResNet-50 (see results in Appendix Table 7). This practice maintains compatibility with a wide range of UDA methods and pre-trained features.
> >
> > Thus, our major contributions revolve around **construction of a series of intermediate domains** connecting the source and target domains in the latent feature space. The comprehensive experiments in Tables 2-6 and Appendix Table 7 as well as the visualized adaptation process in Figure 9 substantiate the feasibility and effectiveness of our method in UDA tasks.

---

> > ### Comment · Reviewer_T69j · 2023-11-20
> >
> > Thank you for the response to address my concerns.

---

### Official Review · Reviewer_vJxQ · 2023-10-31

**Soundness:** 3 good
**Presentation:** 3 good
**Contribution:** 3 good
**Rating:** 8
**Confidence:** 3

**Summary:**

This paper introduces Gradual Domain Adaptation via Gradient Flow (GGF) to generate intermediate domains (when they are not given), which are then used for the fine-tuning process of a gradual domain adaptation (GDA) setting. In the proposed GGF method, feature representations from the source domain are gradually transported to the target domain by using a Wasserstein gradient flow (WGF) that aims to minimize the following three designed energies: distribution-based energy, classifier-based energy, and sample-based energy. In addition, the authors provide a theoretical analysis of its generalization bound. Experimental results on several benchmark datasets demonstrate the advantage of the proposed GGF method compared to other current baselines.

**Strengths:**

The paper is well-written and relatively easy to follow. The idea of generating (synthetic) intermediate domains with preserving labels is novel and interesting.  I appreciate the authors' effort to provide a theoretical guarantee of a target error-bound of the proposed GGF.

**Weaknesses:**

1. Regarding the proposed method, can the authors provide an estimation of the sufficient number of the intermediate domains $T$ to obtain a pre-defined target performance? What happens when $T \to \infty$?

2. Current experimental results of the paper are supportive and promising, but not very convincing. In particular,

- Lack the empirical comparison with the very related baseline [A] in both Tables 1 & 2.

- Currently, the experiment results can only show the benefit of GGF when applying for two baselines (including RSDA and Covi). I am curious how GGF can be combined with others mentioned in the paper.

- GGF seems to be a time-consuming. Comparison regarding the running time with other mentioned baselines?

[A]. @inproceedings{wang2022understanding,
  title={Understanding gradual domain adaptation: Improved analysis, optimal path and beyond},
  author={Wang, Haoxiang and Li, Bo and Zhao, Han},
  booktitle={International Conference on Machine Learning},
  pages={22784--22801},
  year={2022},
  organization={PMLR}
}

Clarification:
1. In Fig. 1, the transformation from the source to the target is performed in the latent space, while everything in the paper is about the original data $x$. Could you please clarify that?

2. The proposed method relies on the labeling function $f$ for each domain, while we do not have access to the labels of target samples in a general UDA framework. Could you please comment on this?

3. Could you please double-check the last equality in Eq. (5). It seems the integral over $\mu_t$ term was missed.

**Questions:**

Please address my concerns/questions in the Weaknesses part above.

---

> ### Author Response · Authors · 2023-11-19
> **Response to Reviewer vJxQ (1/2)**
>
> We express our gratitude for your thorough review. Below, you will find our response addressing your concerns point by point. If you have any additional questions or require further clarification, please feel free to let us know. Your insights are highly valued.
>
> ### For weakness
>
> #### [W.1] Regarding the proposed method, can the authors provide an estimation of the sufficient number of the intermediate domains T to obtain a pre-defined target performance? What happens when T →  $\infty$?
> > Both our theoretical and empirical results unanimously advocate for an appropriate value of $T$, which should be large but not going to $\infty$.
> > - **Theoretically,** Theorem 1 establishes three terms related to $T$ that influence the upper bound. Among them,
> >     - the transport cost $\eta\alpha TU$ increases with $T$;
> >     - the coefficient for the distance between the source and target domains, i.e., $(1-\eta m)^{\alpha T}$, decreases with $T$;
> >     - the last term $\mathcal{O}\left(\frac{\rho B + \sqrt{\log (1 / \delta)}}{\sqrt{n}} T \right)$ increase with $T$.
> >
> >     Thus, a clear trade-off emerges where $T$ **strikes a balance that reduces the upper generalization bound**. The intuition behind this trade-off is that a larger $T$ contributes to reducing distribution differences between adjacent intermediate domains, while an excessively large $T$ may lead to more fine-tuning steps, resulting in overfitting of the classifier.
> > - **Empirically,** a smaller step size $\eta$ or a lower frequency $\alpha$ of generating intermediate domains usually results in a higher number of intermediate domains $T$. As shown in Figure 13, if $\eta$ or $\alpha$ is set too low or too high, meaning that $T$ is either too large or too small, accuracy tends to be sub-optimal. For datasets with large domain gaps, such as MNIST 60° and Office-Home, we observe that  **64 intermediate domains are sufficient to achieve the highest accuracy**.
>
> #### [W.2] Current experimental results of the paper are supportive and promising, but not very convincing.
>
> ##### [W.2a] Lack the empirical comparison with the very related baseline [A] in both Tables 1 & 2.
>
> >This study [A] extends the theoretical findings of [r1], demonstrating that the generalization bound of the **gradual self-training (GST) algorithm in GDA** depends linearly and additively on the parameter $T$. We would like to highlight the difference between  this baseline [A] and ours/GOAT/CNF in that **GST [A] relies on existing intermediate domains**.
> > - Empirical comparison on **GDA tasks**: we implement GST to leverage 1/2/4/8 intermediate domains accessible in the GDA benchmarks; however, we reiterate that the **comparison may be considered unfair** since our method, along with GOAT and CNF, does not require any intermediate domains. The results shown below and also updated into Table 1 in the revised paper provide compelling evidence that
> >    - Ours **significantly outperforms GST in real-world benchmarks with oftentimes low-quality intermediate domains**, e.g., Portraits whose imperfect alignment has been elucidated in Figure 11.
> >    - Even for the synthetic MNIST dataset where real intermediate domains precisely follow the geodesic path, the proposed GGF approaches the accuracy of GST when employing 4 uniformly spaced intermediate domains.
> >
> >
> >   | Dataset \ # Given domains | 1     | 2     | 4     | 8         | GGF (Ours) |
> >   | ------------------------- | ----- | ----- | ----- | --------- | ---------- |
> >   | Portraits                 | 72.85 | 76.55 | 81.45 | 83.75     | **86.16**  |
> >   | MNIST 45°                 | 53.37 | 62.35 | 66.45 | **75.05** | 67.72      |
> >   | MNIST 60°                 | 39.95 | 48.79 | 56.87 | **64.20** | 54.21      |
> >
> > - Empirical comparison on **UDA tasks**: GST [A] is not feasible for comparison in Table 2, due to the absence of intermediate domain tasks in Office-Home. We highlight the **broader applicability** of the proposed GGF in addressing UDA  scenarios that lack access to intermediate domains.
> >
> >[r1] Understanding self-training for gradual domain adaptation, ICML 2020
>
> ##### [W.2b] How GGF can be combined with more baselines mentioned in the paper.
>
> > We would like first to highlight that improving the generalization of pre-trained features by any feature extractor, which is readily accessible nowadays, to target domains constitutes our core motivation and contribution.  As outlined in Appendix D.3, we have extended our method by combining it with additional pre-trained feature extractors on Office-Home, including MSTN and ResNet-50. We re-list the improvements of our results below, indicating **almost consistent improvements** across all tasks.

---

> > ### Comment · Reviewer_vJxQ · 2023-11-22
> > **Thanks for the detailed responses**
> >
> > I would like to thank the author for their comprehensive reply, especially for the additional experimental results. It would be great if the authors could also address the remained questions about clarification.
> >
> > Bests

---

> > > ### Author Response · Authors · 2023-11-22
> > > **Gratitude to Reviewer vJxQ and a gentle reminder for the second part of our response**
> > >
> > > Thank you for your appreciation of our comprehensive reply and the additional experimental results. We would like to assure you that we have indeed addressed the remaining questions and provided further clarification in the second part of our response, **titled "Response to Reviewer vJxQ (2/2)" in the thread** . If there are any specific points you would like further elaboration on or if you have additional questions, please feel free to let us know. We appreciate your feedback and are committed to ensuring clarity and completeness in our responses.

---

> ### Author Response · Authors · 2023-11-19
> **Response to Reviewer vJxQ (2/2)**
>
> ##### Table of response to [W.2b]
>
> > |                | Ar→Cl | Ar→Pr | Ar→Rw | Cl→Ar | Cl→Pr | Cl→Rw | Pr→Ar | Pr→Cl | Pr→Rw | Rw→Ar | Rw→Cl | Rw→Pr | Average |
> > | -------------- | ----- | ----- | ----- | ----- | ----- | ----- | ----- | ----- | ----- | ----- | ----- | ----- | ------- |
> > | Resnet50 + GGF | +2.3  | +2.5  | +2.7  | +2.8  | +1.9  | +1.7  | +4.5  | +1.9  | +2.5  | +2.2  | +2.2  | +1.9  | +2.5    |
> > | MSTN + GGF     | +0.7  | +5.0  | +2.7  | +6.1  | +4.7  | +5.9  | +4.2  | +2.3  | +1.9  | +3.6  | +5.6  | +4.2  | +3.9    |
> > | RSDA + GGF     | +6.9  | +0.2  | +0.9  | +2.0  | +4.3  | +0.7  | -0.1  | +6.7  | +0.5  | 0     | +3.2  | -0.2  | +2.2    |
> > | Covi + GGF     | +0.7  | +0.9  | +0.4  | +1.2  | -0.1  | +0.9  | +0.4  | +1.4  | +1.0  | -0.4  | -0.9  | 0     | +0.5    |
> >
>
> ##### [W.2c] Comparison regarding the running time with other mentioned baselines?
>
> > Our approach employs an efficient forward scheme to discretize Wasserstein gradient flow, whose complexity linearly depends on the number of intermediate domains and the dataset scale.  **Consequently, our time consumption for generating intermediate domains and updating the classifier is comparable to the other two methods.** We provide supportive experiments, including a comparison of running time (s) on the GDA datasets under **16** intermediate domains. The results are presented in the table below and further detailed in Table 8 in Appendix D.4 in the revised paper.
> >
> > We also include Table 1 here for convenient reference. It is noteworthy that while GGF takes longer to train than GOAT, the performance improvement by GGF achieves as high as $18\%$. Considering that GGF converges typically within 15 minutes for each Office-Home task, we assert that GGF well balances between effectiveness and efficiency.
> >
> > |           | GOAT     | CNF   | GGF (Ours) |
> > | --------- | -------- | ----- | ---------- |
> > | Portraits | **2.84** | 5.25  | 5.76       |
> > | MNIST 45°/60° | **7.95** | 14.23 | 29.53  |
> >
> >
> > |           | GOAT     | CNF   | GGF (Ours) |
> > | --------- | -------- | ----- | ---------- |
> > | Portraits | 83.17 | 84.57  | **86.16**     |
> > | MNIST 45° | 61.45 | 62.55 | **67.72**      |
> > | MNIST 60° | 46.19 | 41.18 | **54.21**      |
> >
>
>
> ### For clarification
>
> #### [C.1] In Figure 1, the transformation from the source to the target is performed in the latent space, while everything in the paper is about the original data x. Could you please clarify that?
>
> > We apologize for any confusion. As explained in Section 5.1 of the experiment setting, the transformation is performed in the (latent) feature space. However, in the methodology and theory sections, we use "$\mathbf{x}$" to refer to both data and latent spaces, which increases the framework's versatility and comprehensive applicability. We appreciate your valuable comments and have already clarified this aspect in the revised manuscript.
>
> #### [C.2] The proposed method relies on the labeling function for each domain, while we do not have access to the labels of target samples in a general UDA framework. Could you please comment on this?
>
> > - There does exist an underlying ground-truth labeling function for each domain, although it is inaccessible. Therefore, it is reasonable to theoretically define them (without accessing them) **for theoretical analysis**.
> > - In this paper, we consider UDA where the target samples lack labels. We generate multiple intermediate domains by gradually transforming labeled source features to the target domain, all while **retaining the original sample labels**. Consequently, we can fine-tune the classifier progressively based on "labelled" intermediate domains.
> >
>
> #### [C.3] Could you please double-check the last equality in Eq. (5). It seems the integral over $\mu_{t}$ term was missed.
>
> > We apologize if there has been any confusion, but we want to clarify that this is not a mistake. The $\mathbf {u} _t$ represents the (velocity) vector field as a function of $\mathbf {x}$, and the complete form of Eq. (5) is given by:
> >
> > $$
> > \mathbf {u}_t (\mathbf{x}) = -\nabla _ \mathbf{x} \log \mu _{t} ( \mathbf{x} ) + \nabla _ \mathbf{x} \log \pi (\mathbf{x})
> > $$
> > Concerning the gradient computation in Wasserstein space $W_2$, as expressed in Eq. (4):
> >
> > $$
> > \mathbf {u}_t = - \nabla _{W _2} \mathcal{E}(\mu _t) = -\nabla \mathcal{E}^\prime(\mu _t)
> > $$
> >
> > This involves taking the **derivative** of $\mathcal{E}(\mu_t)$ first, followed by computing the **gradient** with respect to $\mathbf{x}$. Therefore, for $\mathrm{KL}(\mu_t | \pi)$, the initial derivative step nullifies the integral over $\mu_t$.  We hope this explanation addresses your concerns.

---

> ### Comment · Reviewer_vJxQ · 2023-11-23
>
> Thanks for pointing out that. My apology for not reading your response completely. Most of my concerns have been addressed and I am going to upgrade the score to 8.

---

### Official Review · Reviewer_KhtB · 2023-11-01

**Soundness:** 3 good
**Presentation:** 3 good
**Contribution:** 3 good
**Rating:** 6
**Confidence:** 2

**Summary:**

This paper studies the gradual domain adaptation problem. It leverages gradient flow analysis for generating intermediate domains between the source and target domain to facilitate the adaptation. The proposed approach consists of three main components: (i) A distribution based loss function that transports source features to target ones. (ii) A classifier based regularizer to preserve the label information while transporting the features. (iii) sample based noise reduction regularizer. The proposed approach is supported both theoretically and experimentally where performance gains are consistent.

**Strengths:**

The main strengths of this work are:

(1) The paper is fairly well-written.

(2) The proposed approach is novel and theoretically supported.

(3) The experiments support the effectiveness of the proposed approach.

**Weaknesses:**

While I am not an expert in this field, I would recommend/suggest the following for improving the paper:

1) The methodology section needs some elaboration for better readability. How are the score networks trained? The same question goes to the rectified flow $\nu_\theta$.

2) The theoretical analysis, while supporting the proposed approach, seem not related to the experiments conducted in section 5. I would recommend investing some space in the main paper for implementation details and computational burdens.

3) Regarding the experiments: The following experiments are missing from the main work:

(3a) Hyperparameter ablations: While Table 3 shows the effectiveness of each component of the proposed approach, sensitivity analysis with regard to $\eta_1, \eta_2, \eta_3, \lambda$ are missing.

(3b) While GGF showed enhanced performance when combined with two approaches (Cove and RSDA), it is important to compare different methods in terms of runtime and computational overhead.

(3c) Experiments on more challenging datasets such as DomainNet and ImageNet-C where domain shifts are more severe.

**Questions:**

In addition to the points raised in the weaknesses, I have the following question:

1) From the methodology part, can you explain how is the combined approach in section 3.1 and 3.2 related to the Classifier-Free Diffusion Guidance [A]?

[A] Classifier-Free Diffusion Guidance, NeurIPSW 2021.

---

> ### Author Response · Authors · 2023-11-19
> **Response to Reviewer KhtB (1/3)**
>
> We express our sincere gratitude for your thorough review and valuable insights. Having given careful consideration to your comments, we have made the necessary revisions. If you have any additional concerns or suggestions, we would greatly appreciate hearing them.
>
> #### [W.1] The methodology section needs some elaboration for better readability. E.g. How are the score network and rectified flow trained?
>
> > - We train the two models directly using the **optimization objectives stated in Eq. (7) and Eq. (10)** as the loss functions.
> >
> >   - For the matching term $\nabla_{\tilde{x}} \log  q_\sigma(\mathbf{\tilde{x}}| \mathbf{x})$ in Eq. (7), corrupted examples $\mathbf{\tilde{x}}=\mathbf{x} + \mathbf \epsilon$ are generated based on clean examples, where $\mathbf \epsilon \sim N(\mathbf 0, \sigma^2 \mathbf I)$, resulting in $\nabla_{\tilde{x}} \log  q_\sigma(\mathbf{\tilde{x}}| \mathbf{x}) = - \frac {\mathbf{\tilde{x}}-\mathbf{x}}{\sigma^2}$.
> >   - For the flow matching term in Eq. (10), the conditional vector field is calculated as $v(\mathbf{x} | \mathbf{x}_\tau) = {\mathbf{x}}_1- \mathbf{x}_0$.
> >
> >   Note that Both $\mathbf{x}$ and $\mathbf{x}_1$ are randomly sampled from the target domain, while $\mathbf{x}_0$ is sampled from the source domain.
> > - As detailed in Appendix E, we implement denoise score matching in Eq. (7) for training the score network and flow matching in Eq. (10) for training the Rectified flow **using their officially released codes**.
>
> #### [W.2] The theoretical analysis, while supporting the proposed approach, seems not related to the experiments conducted in Section 5. I would recommend investing some space in the main paper for implementation details and computational burdens.
>
> > - We thank the reviewer for acknowledging our theoretical analysis, that is, providing **the first generalization bound for UDA via constructed intermediate domains and progressive classifier fine-tuning**.
> > - We humbly highlight the **connections of our theoretical analysis to our experiments** in Section 5, including
> >    - **theoretical guarantee for improved performance**: Theorem 1 establishes a tighter upper bound for UDA for even two more distant domains. This is achieved by reducing the coefficient for $W_2\left(\mu_0, \pi\right)$. This justifies the superior performance of the proposed GGF across all three datasets in varying distances between domains.
> >    - **insights for selection of hyperparameters** $\eta_1$, $\alpha$, and $T$:  Due to the monotonicity of the upper bound, the optimal value for the
> remaining hyperparameter after fixing two of them is deterministic, which aligns with the lowest upper bound. We also detail this selection process in  Appendix C.
> >
> > - We have **followed the reviewer's suggestion** by relocating the Definition and Remark on "Continuous Discriminability" for theoretical analysis to the Appendix. Meantime, we have integrated the implementation details mentioned in [W.1], the discussion of computational burdens from [W.3b], and the ablation experiments presented in [W.3a] into the main text of the revised paper.

---

> ### Author Response · Authors · 2023-11-19
> **Response to Reviewer KhtB (2/3)**
>
> #### [W.3] The following experiments are missing from the main work.
>
> ##### [W.3a] Sensitivity analysis about $\eta_1$, $\eta_2$, $\eta_3$, and $\lambda$ are missing.
> > We have taken into account the insightful suggestion from the reviewer and conducted a sensitivity analysis for $\eta_1$, $\eta_2$, $\eta_3$, and $\lambda$ on the Portraits dataset. The results, along with discussions, are presented in the subsequent tables for your convenience. Additionally, we have incorporated these findings into Section 5.2 in our updated manuscript.
> > - **For the step sizes $\eta_1$, $\eta_2$, $\eta_3$ of the three energies,** we re-scale their default values (0.03, 0.08, 0.01 as reported in Table 9) for each while keeping the others fixed. For example, a re-scaling ratio of 75\% for $\eta_1$ yields its value of 0.03 * 75\% = 0.0225.
> >    - The results demonstrate the **insensitivity of the three step sizes** across quite a wide range. Only when $\eta_1$ is reduced to below $0.03\*50$ \% or ${\eta}_2$ is increased to above $0.08\*150$ % does a significant decrease in performance occurs. We attribute this to the larger velocity components induced by the dominating classifier-based energy, which tends to push samples away from the decision boundary. Such behavior contradicts the fundamental principles of GGF, which seeks to leverage gradient flow to influence the decision boundary positively.
> >   - Fortunately, as mentioned in the response to [W.2], our theoretical results established in Theorem 1 offer valuable insights into avoiding solutions where the classifier-based energy dominates. Such solutions, as indicated by our theory, do not contribute to a reduction in the Wasserstein distance between the intermediate to the target domain.
> >
> >   | Re-scaling ratio             | 25%   | 50%   | 75%       | 100%      | 150%  | 200%  | 400%  |
> >   | ------------------------- | ----- | ----- | --------- | --------- | ----- | ----- | ----- |
> >   | $\eta_1=0.03$, fix others | 79.00 | 79.15 | **86.45** | 86.35     | 84.85 | 84.75 | 84.10 |
> >   | $\eta_2=0.08$, fix others | 84.75 | 85.80 | 86.15     | **86.35** | 74.45 | 78.90 | 68.00 |
> >   | $\eta_3=0.01$, fix others | 85.85 | 86.05 | **86.50** | 86.35     | 86.40 | 86.15 | 85.80 |
> >
> >
> > - **For the balance term $\lambda$ between the two classifier-based energies,** we uniformly sample values from [0, 1]. The results below affirm the insensitivity of $\lambda$, as long as it remains below the threshold of 0.6. The reason explaining the decrease in accuracy when $\lambda$ increases is similar to the impact of increasing $\eta_2$ discussed above.
> >
> > 	| $\lambda$                                               | 0         | 0.2   | 0.4   | 0.6   | 0.8   | 1        |
> >   | ------------------------------------------------------- | --------- | ----- | ----- | ----- | ----- | -------- |
> >   | Accuracy                                                | **86.35** | 86.25 | 86.05 | 74.80 | 73.40 | 73.90    |
> >   | Mean magnitude of velocity vectors ($\times 10^{-5}$)   | **4.13**  | 3.73  | 3.32  | 2.94  | 2.60  | 2.61     |
> >   | Median magnitude of velocity vectors ($\times 10^{-5}$) | 0.31      | 0.34  | 0.44  | 0.57  | 0.72  | **0.88** |
> >
>
> ##### [W.3b] It is important to compare different methods in terms of runtime and computational overhead.
>
> > As mentioned in the implementation details (Section 5.1), GGF leverages **off-the-shelf pre-extracted features** from two approaches (Covi and RSDA). Therefore, the extra runtime required by GGF consists of two parts corresponding to the two stages:
> >
> > - In the first stage, we train the score network and rectified flow, with the required time depending on the number of training epochs, which takes approximately 15 minutes for each Office-Home task.
> > - In the second stage, we generate intermediate domains and gradually fine-tune the classifier. Because of the efficient forward scheme, for each task, constructing each intermediate domain takes less than 1 second, and the overall time for fine-tuning the classifier with 16 intermediate domains is less than 1 minute.
> >
> > Hence, **our method not only achieves superior results in UDA tasks but also does so with a comparable time consumption**, compared to the running time of CoVi (about 50 minutes for each task with 100 epochs).  The efficiency of our approach is further supported by a comparison of the time required for the second stage between our method and baseline approaches.  **Table 8** shows that **the overall time consumption of our method aligns favorably with the baseline**.

---

> ### Author Response · Authors · 2023-11-19
> **Response to Reviewer KhtB (3/3)**
>
> ##### [W.3c] Experiments on more challenging datasets such as DomainNet and ImageNet-C where domain shifts are more severe.
>
> > In Appendix D.2, we conduct an additional experiment on the challenging large-scale dataset VisDA-2017 to further validate the effectiveness of our algorithm. The source domain of VisDA comprises 152,397 synthetic images, and the target domain includes 72,372 real-world images across 12 categories. We utilize the state-of-the-art RSDA method for feature extraction and apply GGF in the feature space. The results (see below) demonstrate an improvement, with an increase of 2.3% in the average accuracy across classes and 1.8% in the overall accuracy.
> >
> > | Method     | Acc      |
> > | ---------- | -------- |
> > | CDAN   | 70.0     |
> > | MDD    | 74.6     |
> > | RSDA       | 75.8     |
> > | RSDA + GGF | **77.6** |
> >
>
> #### [Q.1] From the methodology part, can you explain how is the combined approach in Sections 3.1 and 3.2 related to the Classifier-Free Diffusion Guidance [A]?
> >  We explain the connections and differences of our proposed method (when distribution-based energy is KL divergence) with the diffusion model, including both classifier-guided diffusion and classifier-free diffusion in [A]. The intrinsic idea behind all of these methods is the use of Langevin dynamics based on the score $\nabla_\mathbf z\log p(\mathbf z)$. Below, we provide more details to explain that.
> >
> > - **Classifier-guidance diffusion v.s. GGF**
> >
> >    - In [A], classifier guidance incorporates a weighted class-conditional score into the unconditional diffusion, i.e.,
> >     $$
> >     \nabla_\mathbf z\log \tilde p(\mathbf z |c) = \nabla_\mathbf z\log p(\mathbf z) + (w+1) \nabla_\mathbf z\log p_\theta(c|\mathbf z). [\text{E1}]
> >     $$
> >    - For our proposed GGF, when the classifier-based energy is measured by **cross-entropy**, the velocity component can be derived as follows:
> >     $$
> >     \mathbf v_{\text {CE}} = -\nabla_\mathbf z \mathcal L_{\text {CE}}(\mathbf z, h, c) =  \nabla_\mathbf z  (y(\mathbf{z}) \log h(\mathbf{z}; \theta)) = \nabla_\mathbf z   \log p_{\theta}(c|\mathbf{z}).
> >     $$
> >     Here, $y(\mathbf{z})$ represents the one-hot encoding of the label $c$, $h(\mathbf{z}; \theta)$ is the prediction vector of the classifier, and $p_{\theta}(c|\mathbf{z})$ is the component of $h(\mathbf{z}; \theta)$ corresponding to category $c$. Considering that the sampling of distribution-based energy can be expressed as LMC in Eq. (6), the formulation of the combined approach in Sections 3.1 and 3.2 will be:
> >     $$
> >    \mathbf{z}_{t+1}= \mathbf{z}_t - \eta_1 \nabla \log p (\mathbf{z}_t) - \eta_2 \nabla  \log p _{\theta} (c|\mathbf{z}_t) +\sqrt{2 \eta_1} \xi , [\text{E2}]
> >     $$
> >   - Connection: by comparing [E1] and [E2], we observe the seeming similarity of the two approaches, given ours with cross-entropy classification loss.
> >   - Differences: However, **the two approaches differ** in the following aspects.
> >
> >       - Classifier-guidance diffusion is designed to generate samples of a specified category. In contrast, the classifier-based energy employed in our proposed GGF method serves as a penalty term to ensure the preservation of the sample label during feature transformation.
> >       - For generative tasks in classifier-guidance diffusion, the classifier remains fixed. However, in our setting, **the classifier undergoes gradual fine-tuning as the domain changes**.
> >       - Our framework enjoys more flexibility, being readily extendable to use other distribution-based energy metrics, such as MMD and Wasserstein distance, as well as various classifier energies, such as the classifier prediction's entropy.
> >
> >
> > - **Classifier-free diffusion v.s. GGF**
> >
> >    Different from classifier-guidance diffusion, classifier-free diffusion implicitly replaces the classifier $\nabla_\mathbf z\log p(c|\mathbf z)$ by $\nabla_\mathbf z\log p(\mathbf z |c) - \nabla_\mathbf z\log p(\mathbf z)$, resulting in the following equation,
> >     $$
> >     \nabla_\mathbf z\log \tilde p(\mathbf z |c)  = (1+w)  \nabla_\mathbf z\log p(\mathbf z |c) - w \nabla_\mathbf z\log p(\mathbf z),
> >     $$
> >      which requires the labels of the target domain, and differs from our approach that trains the unconditional score function on the **unlabeled target domain**.

---

### Official Review · Reviewer_MZDz · 2023-11-03

**Soundness:** 3 good
**Presentation:** 3 good
**Contribution:** 2 fair
**Rating:** 6
**Confidence:** 4

**Summary:**

The work proposes a gradual domain adaptation method that improves classification results. The method applies Wasserstein gradient flow to minimize a novel energy function. The samples from the source distribution flow gradually to the target distribution by multiple intermediate domains. The experiments show that the classification accuracies are comparable to current classification methods.

**Strengths:**

1. The paper is novel in terms of minimizing an energy function and using that to help a classification task.
2. The writing is clear and the paper is easy to follow.
3. The paper conducts several experiments and numerical comparisons to show that the classification results are comparable with other methods.

**Weaknesses:**

1. In Table 2, it looks like the results are only the best accuracy for three tasks.
2. In Figure 1, it is a bit unclear what the source distribution and target distribution are. Are they both portraits but only the source distribution has labels? It would be more clear if you defne what are $\mu_t, \pi$, such as writing them as metrics based on $x,y$.
3. We recommend the authors cite the following two recent works on MMD and gradient flow:

Fan, J. and Alvarez-Melis, D., 2023. Generating synthetic datasets by interpolating along generalized geodesics. arXiv preprint arXiv:2306.06866.

Hua, X., Nguyen, T., Le, T., Blanchet, J. and Nguyen, V.A., 2023. Dynamic Flows on Curved Space Generated by Labeled Data. arXiv preprint arXiv:2302.00061.

**Questions:**

1. Do you make assumptions about how the distributions look like and if the source and target distributions are close?
2. The accuracies grow as the energy is minimized. Is there a way to measure the quality of the flowed images, both qualitatively or using some metrics like FID?
3. Is it possible to use the Wasserstein distance in the distance-based energy?

---

> ### Author Response · Authors · 2023-11-19
> **Response to Reviewer MZDz (1/2)**
>
> We sincerely thank the reviewer for providing valuable feedback. You may find our response below for your concerns. Please kindly let us know if you have any further concerns.
>
> #### [W.1] In Table 2, it looks like the results are only the best accuracy for three tasks.
>
> > The proposed method, akin to other baselines in Table 2, exhibits fluctuations across different tasks and does not consistently take the lead. However, we highlight that
> >   - our **average performance**, evaluating the overall effectiveness, is unquestionable superior to all state-of-the-art methods;
> >   - towards our primary objective of improving pre-trained features to mitigate distribution shift, our method proves to **consistently outperform 4 backbone pre-trained features in nearly all tasks** as shown in both the subsequent table and Appendix D.3.
> >
> > |                | Ar→Cl | Ar→Pr | Ar→Rw | Cl→Ar | Cl→Pr    | Cl→Rw | Pr→Ar    | Pr→Cl | Pr→Rw | Rw→Ar    | Rw→Cl    | Rw→Pr    | Average |
> >| -------------- | ----- | ----- | ----- | ----- | -------- | ----- | -------- | ----- | ----- | -------- | -------- | -------- | ------- |
> > | Resnet50 + GGF | +2.3 | +2.5 | +2.7 | +2.8 | +1.9 | +1.7 | +4.5 | +1.9 | +2.5 | +2.2 | +2.2 | +1.9 | +2.5 |
> >| MSTN + GGF     | +0.7  | +5.0  | +2.7  | +6.1  | +4.7     | +5.9  | +4.2     | +2.3  | +1.9  | +3.6     | +5.6     | +4.2     | +3.9 |
> > | RSDA + GGF     | +6.9  | +0.2  | +0.9  | +2.0  | +4.3     | +0.7  | -0.1 | +6.7  | +0.5  | 0        | +3.2     | -0.2 | +2.2    |
> >| Covi + GGF     | +0.7  | +0.9  | +0.4  | +1.2  | -0.1 | +0.9  | +0.4     | +1.4  | +1.0  | -0.4 | -0.9 | 0        | +0.5    |
>
> #### [W.2] In Figure 1, it is a bit unclear what the source distribution and target distribution are.
>
> > - The source distribution and target distribution can be described as $\mathbf x_{srs} \sim \mu_0(\mathbf x)$, and $\mathbf x_{tgt} \sim \mu_{T+1}(\mathbf x)$, respectively, with $\mu_t(\mathbf x)$ representing the **marginal probability distribution for the $t$-th domain**. Note that $\pi := \mu_{T+1}$. **Labels are only accessible in the source domain.**
> > - To exemplify, we resort to the GDA benchmark of Portraits established in [r1], where the **source domain** consists of portrait photos from the **1900s** while the **target domain** comprises portraits from the **1970s**. Discrepancies in the distribution between these two domains are acknowledged in the seminal work of [r1].
> > - We apologize for any ambiguity in the illustration and have updated Figure 1 in the revised manuscript.
> >
> > [r1] Understanding self-training for gradual domain adaptation, ICML 2020.
>
> #### [W.3] We recommend the authors cite the following two recent works on MMD and gradient flow.
>
> > Thank you for your valuable suggestions. In our revised manuscript, we have included citations and discussions regarding the works of [r2] and [r3].
> >
> > - Discussion on [r2]
> >   - Problem: the authors considered **transfer learning with a multi-dataset as the source domain**, which differs from ours.
> >   - Methodology: toward the above distinct objective from ours, [r2] generates a synthetic dataset as the source domain through computing **neural optimal transport maps** to map the target domain onto a combination of source domain samples. We have compared our approach with GOAT which also adopts optimal transport in synthesizing intermediate domains, and proved our effectiveness.
> >
> > - Discussion on [r3]
> >   - Problem: the authors proposed to **augment a labeled target domain**, while we consider unlabeled target domains.
> >   - Methodology: the authors proposed to construct a
> > **MMD gradient flow** to generate augmented target samples from source samples. In our response to [Q.3], we also implemented the MMD as a distribution-based energy, and the results demonstrate the superiority of our implementation via Langevin dynamics.
> >
> > [r2] Generating synthetic datasets by interpolating along generalized geodesics, UAI 2023.
> >
> > [r3] Dynamic Flows on Curved Space Generated by Labeled Data, IJCAI 2023.

---

> ### Author Response · Authors · 2023-11-19
> **Response to Reviewer MZDz (2/2)**
>
> #### [Q.1] Do you make assumptions about how the distributions look like and if the source and target distributions are close?
>
> > We refrain from positing explicit assumptions regarding the distance between the source and target distributions. Both our theoretical underpinnings and empirical findings justify that the proposed GGF handles **more distant distributions** than conventional UDA methods.
> >  - **Theoretically**, we juxtapose the generalization bound for conventional UDA utilizing the Wasserstein distance [r4], i.e.,
> >    $$\epsilon_\pi\left(h\right) \leq \epsilon_{\mu_0}\left(h\right)+\rho R W_1\left(\mu_0, \pi\right) +\lambda,$$
> >     which signifies that the distance between the source and target distributions, denoted as $W_1\left(\mu_0, \pi\right)$, should not be too large to make the bound vacuum. In contrast, our theoretical results in Theorem 1 introduce a different coefficient for this distance. This coefficient $(1-\eta m)^{\alpha T}$ decreases as long as the proposed GGF constructs suitable intermediate domains with large sampling steps $\alpha T$, thereby **tightening the upper bound even in cases where the two distributions are distant**. Still, we note that the distance cannot be arbitrarily large.
> >  - **Empirically**, in the dataset of MNIST 60° where the source and target domains are far away, GGF proves to significantly outperform self-training (ST) as a semi-supervised UDA approach. This is achieved through the construction of continuous intermedia domains.
> >
> > [r4] Theoretical analysis of domain adaptation with optimal transport, ECML PKDD, 2017.
>
>
> #### [Q.2] The accuracies grow as the energy is minimized. Is there a way to measure the quality of the flowed images, both qualitatively or using some metrics like FID?
>
> > - **Qualitatively, in Appendix D.5**, we have visualized
> >      - the evolution of generated features by our proposed GGF in Figures 7 and 8, which demonstrates that the generated intermediate domains gradually shift from the source to the target domain;
> >      - the gap between the generated intermediate domains and the real existing intermediate domains in Figures 11 and 12, showcases the high quality of the generated domains that align well with real ones.
> >
> > - **Quantitatively**, we access the quality using the Wasserstein distance and an FID-like metric (defined below) between the features of the generated intermediate domains and those of the target domain; a smaller distance or FID-like score indicates proximity to the target domain. The results presented in the following table validate that as the intermediate domain index increases, **the generated intermediate domains by our GGF are indeed approaching the target.**
> > $$
> > F I D(x, g)=\left\|\mu_x-\mu_g\right\|_2^2+\operatorname{Tr}\left(\Sigma_x+\Sigma_g-2\sqrt{\Sigma_x \Sigma_g}\right)
> > $$
> >    | Intermediate domain index | # 0  | # 5  | # 10 | # 15 | # 20 |
> >    | ------------------------- | ---- | ---- | ---- | ---- | ---- |
> >    | Accuracy (%)              | 74.8 | 75.0 | 78.0 | 82.1 | 86.5 |
> >    | Wasserstein               | 4.96 | 3.28 | 2.33 | 1.71 | 1.24 |
> >    | FID like metric           | 2.37 | 1.47 | 1.07 | 0.81 | 0.62 |
> >
>
> #### [Q.3] Is it possible to use the Wasserstein distance in the distance-based energy?
>
> > Yes. Though various metrics or sampling methods can be employed to implement distance-based energy, we prove in the following ablation that our implementation of KL-divergence via Langevin dynamics offers the **advantage in both accuracy and efficiency**.
> >    - We consider 4 alternatives for comparison, including two implementations of Wasserstein distance, Maximum Mean Discrepancy (MMD), and the implementation of KL-divergence via SVGD [r6]. Specifically, we note that the two implementations of Wasserstein gradient flow are by the
> >      - Forward scheme: $\mu_{t+1}=\left(I-\gamma \nabla_{W_2} \mathcal{E}\left(\mu_t\right)\right)_{\\#} \mu_t$, and
> >      - JKO scheme (backward): $\mu_{t+1}=\arg \min_{\mu} \mathcal{E}(\mu)+\frac{1}{2 \gamma} W_2^2\left(\mu, \mu_{t}\right)$, respectively.
> >
> >       Here, $W_2$, and $\gamma$ represent the Wasserstein distance and step size.
> >  - We provide the results in the following table, from which we observe that the Langevin dynamic we adopt contributes to higher accuracy while requiring less computation time.
> >
> >    |              | Wasserstein (forward) | Wasserstein (backward) | MMD (backward) | KL (SVGD) | Ours (Langevin)|
> >    | ------------ | --------------------- | ---------------------- | -------------- | --------- | --------------- |
> >    | Accuracy (%) | 84.2                  | 84.5                   | 84.4           | 81.9      | **86.5**        |
> >    | Run time (s) | **3.41**              | 70.95                  | 83.66          | 160.75    | 5.76            |
> >
> > [r5] Neural Wasserstein Gradient Flows for Discrepancies with Riesz Kernels, ICML 2023.
> >
> > [r6] Stein Variational Gradient Descent as Gradient Flow, NIPS 2017.

---

### Author Response · Authors · 2023-11-19
**General Response**

Dear Reviewers:

We thank the reviewers for their diligent efforts and high-quality reviews. We have revised our paper to incorporate the valuable feedback provided in the reviews. For your convenience, we have temporarily highlighted these updates in blue. Please find below a summary of the changes we made:

> - The theoretical sections about *Continuous Discriminability* have been relocated from the main text to Appendix B.2.
> - We have added some new experiments, including a comparison with the GST baseline (Table 1), a sensitivity analysis of hyperparameters (Table 4), the results of MSTN+GGF on Office-Home (Table 7), and a comparison of time burdens (Table 8).
> - We have incorporated two new references, as suggested by Reviewer MZDz.
> - We have made several minor updates to our work based on the reviewers' recommendations, including improving the Figure 1 illustration, updating the original energy decomposition in Section 2.2, providing more training details of the score network in Section 3.1, and including two discretization schemes in Appendix A.3.

We sincerely hope that our response and revisions have addressed all concerns raised by the reviewers. Please let us know if our response satisfactorily answers the questions you had for this paper. Thank you once again for your time and effort.

Best regards,

Authors

---

### Meta-Review · Area_Chair_QtFg · 2023-12-05

**Metareview:**

The paper has been reviewed by four expert reviewers. The reviewers have identified several strengths in the paper, including its novelty, theoretical grounding, and experimental results. However, there are also some important concerns and questions that need to be addressed to improve the paper's quality and clarity. Following the unanimous decision of the reviewers, I suggest acceptance and urge authors to fix the remaining issues.

**Justification For Why Not Higher Score:**

The paper clearly has a merit to be published as all reviewers and I agree on acceptance. However, it is still targeting a very small application domain (gradual domain adaptation) and the proposed algorithmic novelty is application specific limiting generality.

**Justification For Why Not Lower Score:**

The paper clearly has a merit to be published as all reviewers and I agree on acceptance. Moreover, the empirical analysis is rigorous and the theoretical study is meaningful to be shared. The paper deserves to be recognized with spotlight distinction.

---

### Decision · Program_Chairs · 2024-01-16

Accept (spotlight)